# Air temperature and precipitation constraining the modelled wetland methane emissions in a boreal region in Northern Europe

Tuula Aalto[1], Aki Tsuruta[1], Jarmo Mäkelä[2], Jurek Müller[3,4], Maria Tenkanen[1], Eleanor Burke[5], Sarah Chadburn[6], Yao Gao[1], Vilma Mannisenaho[1], Thomas Kleinen[7], Hanna Lee[8,9], Antti Leppänen[2], Tiina Markkanen[1], Stefano Materia[10], Paul A. Miller[11], Daniele Peano[10], Olli Peltola[12], Benjamin Poulter[13], Maarit Raivonen[14], Marielle Saunois[15], David Wårlind[11], Sönke Zaehle[16]

[1] Finnish Meteorological Institute, Helsinki, FI-00560, Finland (tuula.aalto@fmi.fi)
[2] CSC Centre of Scientific Computing, Espoo, Finland
[3] Climate and Environmental Physics, Physics Institute, University of Bern, Bern, Switzerland
[4] Oeschger Centre for Climate Change Research, University of Bern, Bern, Switzerland
[5] Met Office Hadley Centre, Exeter, UK
[6] University of Exeter, Exeter, UK
[7] Max-Planck-Institute for Meteorology, Hamburg, Germany
[8] NORCE Norwegian Research Centre AS, Bjerknes Centre for Climate Research, Bergen, Norway
[9] Department of Biology, Norwegian University of Science and Technology, Trondheim, Norway
[10] Fondazione Centro Euro-Mediterraneo sui Cambiamenti Climatici, CSP, Bologna, Italy
[11] Department of Physical Geography and Ecosystem Science, Faculty of Science, Lund University, Sweden
[12] Natural Resources Institute Finland (Luke), Latokartanonkaari 9, Helsinki, 00790, Finland
[13] NASA GSFC, Earth Sciences Division, Biospheric Sciences Laboratory, Greenbelt, USA
[14] Institute for Atmospheric and Earth System Research (INAR)/Physics, Faculty of Science, University of Helsinki, P.O. Box 68, 00014 Helsinki, Finland
[15] Laboratoire des Sciences du Climat et de l'Environnement, LSCE-IPSL (CEA-CNRS-UVSQ), Université Paris-Saclay, France
[16] Max-Planck-Institute for Biogeochemistry, Jena, Germany

*Correspondence to*: Tuula Aalto (tuula.aalto@fmi.fi)

## Abstract

Wetland methane responses to temperature and precipitation are studied in a boreal wetland-rich region in Northern Europe using ecosystem process models. Six ecosystem models (JSBACH-HIMMELI, LPX-Bern, LPJ-GUESS, JULES, CLM4.5 and CLM5) are compared to multi-model means of ecosystem models and atmospheric inversions from the Global Carbon Project and up-scaled eddy covariance flux results for their temperature and precipitation responses and seasonal cycles of the regional fluxes. Two models with contrasting response patterns, LPX-Bern and JSBACH-HIMMELI, are used as priors in atmospheric

inversions with Carbon Tracker Europe – CH4 in order to find out how the assimilation of atmospheric concentration data changes the flux estimates and how this alters the interpretation of the flux responses to temperature and precipitation. Inversion moves wetland emissions of both models towards co-limitation by temperature and precipitation. Between 2000 and 2018, periods of high temperature and/or high precipitation often resulted in increased emissions. However, the dry summer of 2018 did not result in increased emissions despite the high temperatures. The process models show strong temperature as well as strong precipitation responses for the region (51-91% of the variance explained by both). The month with the highest emissions varies from May to September among the models. However, multi-model means, inversions and up-scaled eddy covariance flux observations agree on the month of maximum emissions and are co-limited by temperature and precipitation. The set-up of different emission components (peatland emissions, mineral land fluxes) have an important role in building up the response patterns. Considering the significant differences among the models, it is essential to pay more attention to the regional representation of wet and dry mineral soils and periodic flooding which contribute to the seasonality and magnitude of methane fluxes. The realistic representation of temperature dependence of the peat soil fluxes is also important. Furthermore, it is important to use process-based descriptions for both mineral and peat soil fluxes to simulate the flux responses to climate drivers.

## 1 Introduction

Wetlands are the largest natural source of methane, contributing about 30-40% to the global methane emissions (Saunois et al., 2020, Poulter et al., 2017). Wetlands considered in this study include those on peatlands and mineral lands as well as periodically inundated, i.e. flooded lands. Temperature, soil moisture, water table depth and primary production drive the carbon accumulation, respiration and methane emissions from peatlands and are modelled by ecosystem process models using atmospheric climate data such as temperature and precipitation as input to the simulations. Methane production takes place in water-saturated peat soil layers with limited oxygen availability via anoxic decomposition of soil organic matter by methanogenic microbes. There are accurate peatland maps for the northern regions based on in situ data of peat layer thickness (e.g. Xu et al., 2018, Tanneberger et al., 2017), which enable estimations of the peatland methane emissions by process models if the soil water table level and soil carbon processes providing substrate for methane production are well represented. Land not covered by peatlands includes mineral land. In addition, mineral lands can act as a source of methane if the soil is very moist or inundated (Lohila et al., 2016, Wolf et al., 2011, see also Bansal et al., 2023), with a significant contribution from the organic layer on top of the soil. The soil moisture and land inundation can also be estimated by models together with peat accumulation, though it is still challenging (e.g. Loisel, et al., 2021, Ito et al., 2020). Soil moisture is an important input variable for mineral soil emission modelling (e.g. Curry et al., 2007).

In an attempt to realistically take into account the dynamical changes in total methane emitting area, many process models use wetland extent from remote sensing. However, this feature is badly represented especially in the boreal zone because forests

shadow the inundated areas, and lakes are easily misinterpreted as inundated lands (e.g. Olefeldt et al., 2021, Mahoney et al., 2020, Battaglia et al., 2020, Cohen et al., 2016, Chapman et al., 2016, Papa et al., 2006). Lakes do have methane emissions that may contribute up to one third of boreal biogenic emissions (Guo et al., 2020), but descriptions of lake methane processes are often missing from ecosystem models. Large lakes and rivers have been mapped with high precision, but small ponds, pools, seasonal inundation and low-order streams that may have high methane emissions are challenging to detect accurately (Olefeldt et al., 2021). Permanent water bodies (e.g., lakes, rivers and reservoirs) are usually removed to only cover inundated and non-inundated vegetated wetlands (Zhang et al., 2021). Inundation products are used either as static maps or with inter-annual/ month-to-month variation. As a result, the model predictions of regional annual cycles of methane emission differ and the future estimates of the total global methane emissions are highly variable (Stocker et al., 2013, Saunois et al., 2020). Therefore, instead of studying response to wetland extent, it is useful to take a more climate-oriented perspective to the drivers of the methane emission in order to make better predictions of the responses to future climate change (Koffi et al., 2020). Further, it is important to emphasize the regional approaches as the drivers of emissions vary widely in their spatial distribution, climate and ecosystem type (Stavert et al., 2020). It is important to study the responses of the regional emissions to air temperature and precipitation, as it defines the response of regional wetland emissions to climate change.

Precipitation is the primary environmental driver for soil water dynamics during the growing season, and it can immediately impact the surface soil moisture, while its effect on the water table becomes apparent after a few days or weeks (e.g. Rinne et al., 2020, Gao et al., 2016). Wide-spread periodical inundation, i.e. flooding may appear in spring due to melting snow. In the future, the amount of precipitation is projected to increase in the boreal zone (Putnam et al.,2017, Ruosteenoja et al., 2016). This would potentially lead to wetland expansion (Poulter et al., 2017), although increased evapotranspiration may counteract this (Helbig et al., 2020). Furthermore, rising temperature enhances methanogenesis in the wet soils (Koffi et al., 2020). While wetland extent is the most significant driver of methane emissions in process models (Poulter et al., 2017), soil temperature was shown to be the dominant driver for the inter-annual variability in methane emissions in North America and soil moisture in Western Siberian Lowlands in Russia (Thompson et al, 2016) according to atmospheric inversion modelling and analysis of the results using climate reanalysis data (Dee et al., 2011). Soil moisture was also connected to soil carbon content and methane emissions in Fennoscandia (Albuhaisi et al., 2023), and in Finnish landscape level studies (Räsänen et al., 2021, Vainio et al.,2021).

Atmospheric inverse models rely on atmospheric methane concentrations, and they provide a top-down view of the methane emissions. Their results can be used to study responses of regional methane emissions to climate drivers. In process models the responses are more subject to how the processes were built and dependencies constructed, and how the fluxes were up-scaled. Therefore, it is worthwhile comparing the atmospheric inversion models to process models and study how their regional emission estimates and climate responses differ. Here we compare temperature and precipitation responses from ecosystem process models participating in the H2020-CRESCENDO project for model development. We compare their results to the

ensemble of models from the Global Carbon Project (GCP) 2020 estimation of the global methane budget (Saunois et al., 2020). We use two of the models as well as the average of the GCP land ecosystem model ensemble (Saunois et al., 2020, Poulter et al., 2017) as priors of wetland emissions to inversions with Carbon Tracker Europe – CH4 (Tsuruta et al., 2017).

We determine the sensitivity of the inversion to its prior and how this changes the interpretation of the flux responses to precipitation and temperature change in the boreal region in Fennoscandia. As a result, we obtain an assessment for process-based models using atmospheric inversion modelling, providing guidance on how to improve their climate responses. We get an estimate of how the temperature and precipitation responses vary between the process models and how the extreme climate conditions of the recent years are reflected in the methane emissions.

## 2.    Materials and methods

The ecosystem process models are introduced here together with the inversion system, observations and other materials used in this study. Of the ecosystem models, the wetland descriptions of JSBACH-HIMMELI, LPJ-GUESS, JULES, CLM4.5 and CLM5 were further developed in the recent H2020-CRESCENDO project and results of the standalone simulations made for

the project are used here. CLM5 and JULES results from the coupled Earth System Model simulations were also retrieved from the recent coupled model inter-comparison project phase 6 (CMIP6, Eyring et al., 2016) data archive, and utilized in the work. We also include the LPX-Bern v.1.4 model which participated in Global Carbon Project (GCP) 2020 estimation of global methane budget (Saunois et al., 2020). Further, the ensemble mean of 12 ecosystem models from GCP is used for comparisons with the individual models, as well as the GCP ensemble mean of atmospheric inversions. The ensemble mean

of the GCP ecosystem models, as well as two ecosystem models with contrasting responses (JSBACH-HIMMELI and LPX-Bern), were used as priors in atmospheric inversions with Carbon Tracker Europe – CH4. Models and simulation set-ups are briefly introduced below and in Table 1.

### 2.1 JSBACH-HIMMELI

The ecosystem process model JSBACH version 3.2 with HIMMELI methane module version 1.0 (hereafter called JSBACH-

125 H) was applied in this work. JSBACH is the land component of Max Planck Institute Earth System Model (MPI-ESM) version 1.2 (Mauritsen et al, 2019) and includes a multilayer hydrology model (Hagemann and Stacke, 2015) and representation of soil carbon by YASSO model (Goll et al., 2015). The HIMMELI model, coupled to JSBACH, describes the emission of methane from peatlands (Raivonen et al., 2017), including production, oxidation, diffusion, plant transport and ebullition processes in a multi-layer wetland scheme. For soil organic matter (SOM) decomposition, JSBACH employs the soil carbon

model YASSO (Tuomi et al., 2009; Goll et al., 2015). The specific conditions of peatlands were taken into account in YASSO, following the approach in the peatland carbon model for LPJ (Kleinen et al., 2012) and JSBACH peatland implementation in Kleinen et al. (2020). YASSO uses four C pools for leaf and woody litter, representing the carbon fractions soluble in acid (A), water (W), and ethanol (E), as well as a nonsoluble (N) fraction. A fifth carbon pool is a humus pool containing SOM that

has already undergone substantial decomposition. For the application to peatlands, the humus pool was modified to represent a catotelm carbon pool containing the carbon in the permanently anoxic part of the soil column. For anaerobic decomposition in the acrotelm, a fraction of the soil column was determined that was below the current water table. Decomposition rates were reduced in this part of the soil column by multiplying decomposition rate constants for all C pools with a modification factor $\eta_{anox}=0.35$, following Wania et al. (2010). For the peatland-specific decomposition in the acrotelm, the relative mass flow magnitudes from nonsoluble to acid-soluble were reduced from $p_{3,nonpeat}=0.83$ in the original formulation to $p_{3,peat}=0.66$ for the peatland case. Furthermore, the mass flow magnitude from the nonsoluble to the catotelm C pool was set to $p_{N2cato}=0.17$ for improved peat accumulation rate. The water table level is simulated using a TOPMODEL approach (Kleinen et al., 2020) and the substrate for methane production is received from JSBACH soil anoxic respiration. Other versions of JSBACH-H have been lately developed for studying drained peatland forest management options (Tyystjärvi et al., 2024, in discussion, preprint available, Li et al., 2024, in print, pre-proof available). Methane fluxes in mineral lands are driven by soil moisture from JSBACH hydrology model. The wet mineral soil emissions depend on the soil heterotrophic respiration from JSBACH and a soil moisture threshold is applied for the emissions using approach by Spahni et al (2011). Soil sink for methane is calculated using a model by Curry et al. (2007) for methane diffusion and oxidation in dry soils. JSBACH-H was run at 0.1° resolution over the domain (Fig. 1), with land cover from EU-CORINE interpreting bogs and inland freshwater marshes as methane emitting peatlands (HIMMELI approach) and all other lands as mineral lands. For reference, global runs were also made with 1.875° resolution following the GCP protocol (see Sect. 1.1.8).

## 2.2 LPX-Bern

The Land surface Processes and eXchanges (LPX-Bern) model version 1.4 (Lienert and Joos, 2018, Stocker 2013, Spahni et al., 2013, Spahni et al., 2011) is a dynamic global vegetation model. The vegetation composition for a given land-use class is determined dynamically, allowing the different plant functional types to compete for resources. The configuration with DYPTOP (Dynamical Peatland Model Based on TOPMODEL) combines an inundation model with a model that determines peatland growth conditions to simulate the peatland spatial distribution and temporal changes. DYPTOP accounts for the feedback between inundation dynamics, regional hydrology and peatland establishment, and estimates the distribution of peatlands versus mineral lands. The LPX-Bern model simulates peatland-specific soil carbon dynamics informed by water table position and peatland specific vegetation classes (Sphagnum, Graminoids relevant for the boreal zone), and interaction of the carbon and nitrogen cycles. Methane production, oxidation and transport processes are calculated according to Wania et al., (2010). Model runs were originally made with 0.5° resolution, following the GCP protocol (see Sect. 1.1.8).

## 2.3 LPJ-GUESS

The Lund-Potsdam-Jena General Ecosystem Simulator version 4.0 (LPJ-GUESS, Lindeskog et al., 2013, Smith et al., 2014) with methane module WHyMe (Wania et al, 2010) is a process-based dynamic vegetation and biogeochemistry model and the terrestrial biosphere component in the European community Earth-System Model (EC-Earth-Veg, Hazeleger and Bitanja.

2012, Döscher et al. 2021). LPJ-GUESS land use is described by the Land Use Harmonization version 2 (Hurtt et al, 2020). WHyMe simulates methane production, three pathways of methane transport (diffusion, plant-mediated transport and ebullition) and methane oxidation. LPJ-GUESS-WHyMe stand-alone simulations for CRESCENDO project were made using a prescribed peatland map at a 0.5° resolution.

## 2.4 CLM

The Community Land Model (CLM) is the land component of the Community Earth System Model (CESM). CLM uses the biogeochemical configuration of Biome-BGC (Koven et al., 2013). CLM version 4.5 (Oleson et al., 2013) is the land component of CESM version 1.2 and of the CMCC Coupled Model version 2 (CMCC-CM2, Cherchi et al, 2019) ) and CMCC Earth System Model version 2 (CMCC-ESM2, Lovato et al, 2022). CLM version 5.0 (Lawrence et al., 2019) is the land component of CESM version 2 (Danabasoglu et al, 2020) and of the Norwegian Earth System Model 2 (NORESM2, Seland et al., 2020). CLM4.5 and CLM5 differ e.g. in their description of nutrient dynamics, hydrology parameterization, root profile, nitrogen cycling, and phenology (Lawrence et al., 2019); a new feature in CLM5 is e.g. rain threshold for growth of deciduous vegetation (Peano et al., 2021). The methane emission scheme in CLM includes production, oxidation, ebullition, diffusion and plant transport processes in several soil layers (Meng et al., 2012, Riley et al., 2011). Methane production in the soil layers is calculated as a fraction of aerobic respiration and takes into account e.g. soil pH. Aerobic respiration depends on soil temperature, carbon content and soil moisture. The methane oxidation rate is co-limited by oxygen concentration and methane concentration. The total emissions of a grid cell are calculated for the land area that is considered water-saturated. The saturated and unsaturated grid cell area fractions are determined according to a topographic index approach. In addition, the model takes into explicit account multiple processes during e.g. melting period and thus the saturated fraction can vary largely over the growing season, affecting the methane emissions.

CLM4.5 simulations were originally made at 1.25 x 0.9375° resolution and CLM5 at a 0.5° resolution. Results of the CLM5/ NorESM2-LM coupled simulation from the CMIP6 data archive (https://esgf-node.llnl.gov/projects/cmip6/) are also used in this study.

## 2.5 JULES

JULES-ES version 1.0 (JULES) is the Earth System configuration of the Joint-UK Land Environment Simulator, and land component of the UK community Earth System Model UKESM1 (Sellar et al., 2019). The wetland methane emission in JULES is calculated from soil temperature and substrate availability, and this is then multiplied by grid box saturated fraction (calculated using a topographic index approach) to give the grid box methane emissions (Gedney et al., 2004). Recently, the scheme was updated to calculate methane production on multiple vertical soil layers (Comyn-Platt et al., 2018). It also includes

an empirical decay factor for oxidation (see Chadburn et al., 2020). Wetlands JULES stand-alone simulations were made at a 0.5° x 0.5° resolution. Results of the JULES-ES/ UKESM1-0-LLES-ES-1.0, coupled simulation from the CMIP6 data archive (https://esgf-node.llnl.gov/projects/cmip6/) are also used in this study.

## 2.6 Global Carbon Project models

The Global Carbon Project (GCP) effort for assessing global methane emissions (Saunois et al., 2020) included contributions from ecosystem models and atmospheric inversion models. The land surface model simulations followed a protocol (Saunois et al., 2020), where the models were run with prescribed remote-sensing based year-to year varying wetland area and dynamics dataset WAD2M (Wetland Area Dynamics for Methane Modeling, Zhang et al., 2021) and common climate drivers. This ensemble, of which we used data from 12 models (ELM, DLEM, TEM_MDM, TRIPLEX-GHG, JSBACH, JULES, LPJ-MPI, LPJ-WSL, LPJ-GUESS, LPX-Bern, ORCHIDEE, VISIT), is hereafter referred as GCP-diag. We also used data from model runs where the models used their own approaches to simulate the wetland distributions. This ensemble mean of 8 models (ELM, JSBACH, JULES, LPJ-MPI, LPJ-WSL, LPX-Bern, ORCHIDEE, VISIT), is below referred as GCP-prog. The GCP effort included atmospheric inversion model simulations to provide a top-down view of emissions informed by atmospheric concentration observations. Here we used a mean of five inversion frameworks (TM5-4DVAR, NIES-TM, NICAM-TM 4D-VAR, GELCA, CTE – CH4, set-ups described in Saunois et al., 2020) for comparing the seasonal cycle of process model wetland emissions to seasonality from inversions. Wetland methane fluxes were extracted from the flux totals by the participating research groups and the wetland proportion of the total flux thus depends on the individual approaches chosen, and on the priors used. The wetland priors for the inversions were obtained from different sources. The WETCHIMP ensemble mean (Melton et al., 2013), or e.g. VISIT ecosystem model were used by the inversion models listed above. In the GCP protocol the prior wetland emission information needed for the inversions was obtained from the climatological mean of models from a previous study by Poulter et al., (2017). The GCP-prior from the protocol was used in this work as a prior for Carbon Tracker Europe – CH4 inversions, as well as other priors from LPX-Bern and JSBACH-H (see sect. 1.1.9).

## 2.7 Ecosystem model simulations

The experimental set-ups to run the ecosystem models typically include spin-up by recycling the climate mean and variability from a decadal time period in the beginning of the 20th century, and transient carbon dioxide, climate and land-use runs over several decades until present day. The CRESCENDO models (JULES, CLM4.5, CLM5 and LPJ-GUESS) were run with climate from CRUNCEP version 7 (Viovy et al., 2018) from 1901-2014. For the GCP models and LPX-Bern the simulations covered the period from 1901 through the end of 2018 (GCP 2017), forced by CRU-JRA reconstructed climate fields (Harris, 2019). JSBACH-H was run from 1999 to 2018 with climate from CRU-HARMONIE, and globally with CRU-JRA. Ecosystem process model results were provided on a 0.5 -degree grid (or 0.1 degree for JSBACH-H). The JSBACH-H, LPX-Bern and GCP-prior results were remapped by bilinear interpolation onto 1x1 degree grid for use in CTE-CH4 atmospheric

inversions. Analyses of flux results including ecosystem process model results, atmospheric inversion results and up-scaled eddy fluxes was made on a 1x1 degree grid.

## 2.8 CTE - CH4

Carbon Tracker Europe – CH4 (CTE-CH4) is a data assimilation system that optimizes total global CH4 fluxes (Tsuruta et al., 2017), developed from Carbon Tracker Europe for CO2 (Peters et al., 2005; van der Laan-Luijkx et al., 2017). The system is
based on an ensemble Kalman filter with 500 ensemble members and a fixed lag assimilation window of 5 weeks. Atmospheric methane observation data, mostly surface in situ observations from the OBSPACK v2.0 compilation (Cooperative Global Atmospheric Data Integration Project, 2020), is assimilated into the system. In Northern Europe there were over 10 atmospheric stations that continuously or semi-continuously observed methane between the years 2005 and 2018 (see Fig. 1 and Suppl. Table S1). The TM5 atmospheric chemistry transport model (Krol et al., 2005) is applied to simulate the
atmospheric transport of methane. The runs were forced by the ERA-Interim meteorological reanalysis (Dee et al., 2011). The prior natural surface fluxes which are optimized by CTE-CH4 come from ecosystem process models. Results from the two models introduced above (LPX-Bern and JSBACH-H), and the mean of the GCP models, were used as priors in the inversions. The fluxes are optimized on a 1x1 degree resolution in Europe, but here we studied the sum of emissions from a region in Northern Europe (Fennoscandia), as the posterior fluxes from the inversions were better constrained over that larger region
than 1x1 degree, i.e. the flux uncertainty becomes very large in pixel resolution given the limited number of surface stations. For anthropogenic emissions (optimised separately) we used estimates from EDGAR v5.0 (Crippa et al., 2020), for fire GFED v4.1s (Giglio et al., 2013), and for termites VISIT (Ito et al., 2012), and for oceanic sources we used estimates based on ECMWF data (Tsuruta et al., 2017). The atmospheric sink was calculated using prescribed data obtained from the TM5 chemistry model (e.g. OH fields). The set-up of CTE-CH4 is described in more detail in Tenkanen et al. (2021).

## 2.9 Up-scaled flux observations

The gridded wetland flux product was based on up-scaling of observed (eddy covariance) methane fluxes (Peltola et al., 2019). Fluxes from 25 northern (>45°N) sites were used in constructing random forest models, which consist of a large number of regression trees. Random forest is a machine-learning algorithm that can be used for classification or regression analyses (Breiman, 2001). The random forest model had originally 15 explanatory input variables, e.g. temperature, precipitation,
satellite data of greenness index etc. The up-scaled product was prepared for three wetland maps (LPX-Bern DYPTOP, Stocker et al., 2014, GLWD, Lehner and Döll, 2004, and PEATMAP, Xu et al., 2018). The comparisons were made against the grid-wise mean of the three emission maps available for years 2013 and 2014.

## 2.10 Climate

Meteorological data for studying temperature and precipitation responses was obtained from CRU-JRA. CRU gridded datasets
are found to be suitable for vegetation analyses and well comparable to e.g. MERRA-2 and ERA5-Land reanalysis datasets,

performing well even in remote areas with few observations (Zandler et al., 2020). The CRU-JRA dataset is constructed by re-gridding reanalysis data (JRA, produced by the Japanese Meteorological Agency, Kobayashi et al., 2015), aligned with the CRU TS 4.04 data (Harris 2019, 2020). The CRU-JRA dataset includes (0.5° x 0.5°) gridded 2-m temperature and total precipitation, which are used in this work.


To test the effect of alternative temperature and precipitation data in addition to CRU-JRA, we studied the coupled model runs from the CMIP6 archive, where JULES was coupled with UK-ESM and CLM5 with NOR-ESM2 and thus subject to and interacting with the climate from the coupled model. The results did not change much in terms of placing the highest methane emissions in the temperature-precipitation space. Air temperature explained 76% of the flux variation in JULES and 71% in

the coupled model run and 48% in CLM5 and 37% in the coupled model run. Precipitation explained less than 10% of the flux variation in all model runs (Suppl. Figure S1). To further test the impact of two different bias-corrected data sets, we plotted CRU-HARMONIE and CRU-JRA against JSBACH-H results (Suppl. Figure S2), showing only small differences regarding the response of methane emissions to temperature (temperature explained 82% of the flux variation in both cases) and precipitation (< 5.5%). It was therefore deemed appropriate to use the bias-corrected CRU-JRA data set in our analysis.


The summer months from May to October were examined, as their mean temperatures were always above zero in Fennoscandia (Fig. 2). In May, the soil may still be in freezing or meltwater/inundation state in the northernmost parts of Fennoscandia, but as some of the models above already produce high emissions in that month, we decided to include May in the calculation. The summer months, or length of the thermal growing season varies considerably when moving from south to north in the study

region. In the southernmost parts it is on average >220 days long while in the northernmost parts it is <100 days long (Aalto et al., 2022). Monthly average temperatures were calculated for May to October over the time period from 2000 to 2018. Average precipitation was calculated using precipitation data from the current month and one month before, to include the delay effect in soil water content and thereby correlate better with methane emissions (see e.g. Poulter et al., 2017).

Flux correlations with precipitation and temperature were calculated using the Matlab® statistical package. The proportion of $CH_4$ emission variance explained by temperature (T) and precipitation (P) and both together (TP) were solved using the regress function in the statistical package, performing a least squares fit of flux results on a linear model with temperature and precipitation as predictors (see also Chatterjee et al., 1986).

**3.  Results**

Natural wetland fluxes, including those from peatlands and wet and dry mineral lands as well as inundated lands are studied below for their growing season temperature and precipitation responses in Fennoscandia. Six process models and the mean $CH_4$ emissions from the GCP models are included, as well as results from CTE-CH4 inversions. The results are analysed in

order to examine how the inversions propose to change the prior CH$_4$ emissions and how the correlations with temperature and precipitation, as well as the seasonal cycle of emissions change in the posterior.

## 3.1 Temperature and precipitation responses

The responses of the monthly CH$_4$ emissions to temperature and precipitation varied among the models in Fennoscandia. According to JULES, LPJ-GUESS, JSBACH-H and CLM5, the highest emissions coincided with high temperature (see Fig. 3), while in CLM4.5 the highest emissions resided in the mid-temperature - low precipitation range. In LPX-Bern the highest emissions coincided with high precipitation (see Fig. 3). GCP-diag and GCP-prog had the highest emissions more evenly distributed in the high temperature – high precipitation regime, as could be expected from a mean of several models.

The regressions in Fig. 3 show the correlation of LPJ-GUESS, JSBACH-H and JULES emissions with temperature, indicating that the variance explained was significant as R$^2$ values for temperature were between 0.76 and 0.89 and P-values were $<0.01$. Correlations with precipitation were generally weaker, but still dominated over temperature in LPX-Bern (R$^2_{Precip}$ = 0.50), and CLM4.5 (R$^2_{Precip}$ = 0.47). According to a linear statistical model (Chatterjee et al., 1986) with both temperature and precipitation as predictors, air temperature and precipitation could together explain at maximum 91% of the flux variation (JSBACH-H), but sometimes only 51% (CLM4.5). P-values for the full model were always $< 0.01$. Of the ecosystem models examined, temperature could explain most (R$^2_{Temp}$ = 0.84) of the JSBACH flux variance and precipitation a large part (R$^2_{Precip}$ = 0.50) of the LPX-Bern flux variance. Because of these contrasting features, these two models were chosen as priors in an inversion modelling experiment. The GCP-prior (R$^2_{Temp}$ = 0.45, R$^2_{Precip}$ = 0.35, see Fig. 4) was applied as a third prior for reference.

LPX-Bern, JSBACH-H and GCP-prior were used as prior fluxes in the CTE-CH4 inversions for Fennoscandia. In total, inversions increased the emissions from LPX-Bern priors by 33% and decreased from JSBACH-H priors by 21%, thus bringing the flux estimates closer together. The inversion increased emissions from the GCP prior in the northern parts of Fennoscandia where peatlands are mostly located, and reduced emissions in the southern parts, in total decreasing by 6%. The inversions also increased emissions from the LPX-Bern prior especially in northern Fennoscandia, while there were both decreases and increases from the JSBACH-H prior, with decreases being stronger on average (Suppl Fig S3).

For LPX-Bern the largest posterior increases, i.e. posterior/prior multipliers, were suggested for the high temperature months (Fig. 4). The highest increases from the prior (posterior/prior > 2.0, i.e. above 92 % percentile of all values) occurred above mean monthly temperature of 12.3 °C (64 % percentile of all temperature values, see Fig 4). The highest increase was proposed for July 2014 with second highest mean temperature of 16.7 °C. However, the July 2018 record high heatwave with mean temperature of 17.2 °C was not among the highest posterior increases. The precipitation was record low, only 43 mm in July, which may explain the result, if the prior did not fully capture the possible drought effect and therefore the increase in the

posterior was modest. Some of the highest precipitation months like August 2008, 2016 and July-September 2007 with precipitation exceeding 100 mm, were already above average in the prior emissions but still experienced a large increase in the posterior. This could be because the inversion proposed an increase in the high temperature regime from the prior, and temperatures during these months were above 62% percentile of all values.

For JSBACH-H the largest increases were mostly proposed at the high precipitation regime. The highest increases from the prior (above unity, i.e. above 88% percentile of all values) occurred above 72 mm of precipitation (51% percentile, see Fig. 4). Anomalous high precipitation periods such as those in August 2008 and 2016 and July-September 2007 were significantly increased (> 89% percentile) in the posterior emissions similarly to LPX-Bern. JSBACH-H predicted the largest prior emissions during the warmest months, July 2018 being the highest, followed by July 2014, 2010, 2005 and 2006. In the posterior the fluxes were decreased, but still the emissions stayed above 62% percentile of all values except for 2018, which was close to average. A decrease in soil water table may play a role, as July 2018 suffered from lack of precipitation. The same was true for July 2014 and July 2006, but these months were not clearly distinguishable from precipitation-abundant months July 2005 and July 2010 based on posterior emissions.

In simulations with the GCP-prior the largest increases were appointed to months with highest prior fluxes. The increases were rather scattered over the temperature-precipitation space. In addition, July 2018 did not show high posterior fluxes, aligning with the JSBACH-H results. The highest precipitation months were high also in the posterior. The overall posterior emission pattern followed that of the prior. There was no bias towards high temperature or precipitation regimes, which suggests a balanced prior. The temperature and precipitation correlations of the posterior fluxes were generally weaker than those of the prior for all models (Suppl. Table S2). However, the correlations of the flux multiplier indicated a stronger temperature response in LPX-Bern and a stronger precipitation response in JSBACH-H than in the prior. For the GCP-prior the flux multiplier correlations were weak.

## 3.2 Model components and seasonal cycle

In order to find the reasons behind the specific temperature and precipitation responses, we studied the mean seasonal cycles of the emissions and also the model components which had different seasonal cycles. The total wetland fluxes were summed up from peatland emissions, wet and dry mineral land fluxes as well as emissions from inundated lands. These components cover most of the land area in the study region. However, if we consider wetland area as land area which is wet enough to emit methane, only a fraction of land area is included in the wetland area. Wet mineral land area, inundated land area and peatland area with water table level close to the soil surface all contribute to the total wetland area, and total wetland emissions (see Suppl. Fig. S5-S8).

The peak of the total emissions varied from May (CLM4.5) to September (LPX-Bern), see Fig. 5. The GCP inversion results (mean of five inversion models) peaked during August similarly to the prior. Each model presented differences in seasonal cycle and peak month as they differed in wetland area and model process descriptions. On the contrary, the use of different climate-forcing data products or results from coupled climate simulations as drivers showed minor differences in methane emission responses (Suppl. Fig. S1 and S2). We also studied the seasonal cycle of the up-scaled eddy covariance flux

observations (Fig. 5, see also Suppl. Fig. S9). The up-scaled eddy covariance flux observations (estimated from Peltola et al. 2019 results using mean of the three flux maps) had a broader maximum in July-August, however the temporal extent of the data was quite limited, only two years. Inspection of the model fluxes for the same time period, however, did not reveal significant differences.

JSBACH-H and LPX-Bern were studied more closely because of their contrasting temperature and precipitation dependencies (see 3.1) and differences in seasonal cycles. The LPX-Bern components for peatland fluxes and wet mineral land emissions were largest in magnitude, and comparable to each other in Fennoscandia, but their seasonal cycles were somewhat different (Fig. 6). The soil moisture and consequently the wet mineral land area peaked in autumn (Suppl. Fig. S8) contrary to JSBACH-H (Suppl. Fig. S6), and thus the wet mineral land emissions were at maximum in October in contrast to peatland emissions,

which were at maximum in August. When all components were summed up, an annual cycle was created with maximum wetland emissions in September and October. JSBACH-H wetland emissions were strongly dominated by the peatland component, which had a maximum in July. Wet mineral land component had a broad maximum from August to October, and the dry mineral land sink had a maximum during the warmest summer months (July and August, see Fig 2). All components added together suggested highest emissions in July.

    The CTE-CH4 inversions moved the monthly flux maximum from July to August when JSBACH-H was used as prior in the inversion and from September-October to August-September when LPX-Bern was used as prior (Fig 7). The flux maximum of the GCP-prior was in August and did not change in the posterior. Comparing the change maps for northern Fennoscandia, the inversion with the JSBACH-H and LPX-Bern priors positioned the fluxes to a higher level in August, while in July the

fluxes were placed at a lower level with respect to the seasonal mean adjustment (Suppl Fig S4). This indicates similar changes in the posterior regardless of the prior, i.e. the highest emissions were placed to August. The changes mostly took place in northern peatland areas with high methane emissions.

## 4. Discussion

According to process models, air temperature and precipitation explain a large proportion of the variation in wetland methane emissions from Fennoscandia, which is not surprising given that they comprise major seasonal forcing of the models. Some models (JSBACH-H, LPJ-GUESS and JULES) were clearly more constrained by temperature. The reason behind this

behaviour is linked to strong temperature dependencies in the process descriptions (production, oxidation, transport). According to models and observations, precipitation has a dual role: it presumably increases the wetland area by wetting dry upland soils and raises the water table in the permanent wetlands. In models, weak precipitation constraint could arise from using a constant value or completely neglecting the wet mineral soil emissions or from maintaining static proportions of wet and dry mineral land area over the growing season. For JSBACH-H the reason could possibly be in the temperature dependency of the peatland processes or too small wet mineral land area as opposed to dry land, and in LPJ-GUESS only peatland emissions were included and they had consistently high water table levels. LPX-Bern emissions were strongly constrained by precipitation. In LPX-Bern the wet mineral lands had a large contribution to the emissions, the dynamic wet mineral land area being at largest after prolonged precipitation and generally in autumn (September - October) when the modelled evapotranspiration had already decreased from high growing season levels. According to observations at a boreal site in Hyytiälä, Finland, however, mineral lands were wet and emitting more methane during early season (May- July) than in August – October (Vainio et al., 2021). This suggests that the model should have drier soils in the autumn and the modelled wet mineral soil emissions should have an earlier maximum. In total, the Hyytiälä site always acted as a sink of methane with confined emission patches. The representativity of the finding may be limited since the observations covered only two years, but similar findings have been published earlier (e.g. Kaiser et al., 2018, Warner et al., 2019).

A modeling study by Poulter et al. (2017) concluded that in boreal regions $CH_4$ emissions were best correlated with wetland area, followed by temperature and precipitation (as applied with one-month delay). However, according to Poulter et al. (2017) methane emissions were highly correlated with temperature in some models (e.g. JULES, similarly to our study) which had a high temperature sensitivity. In general, the increased high latitude emissions were consistent with the increase in boreal air temperatures. Sensitivity of the boreal methane emissions to air temperature was confirmed by Koffi et al. (2020), noting that the co-limitation of temperature and precipitation would emerge for the more southern climate zones, whereas in our study co-limitation was found to be present also in boreal zone. According to Figure 2 in Koffi et al. (2020), LPX-Bern was slightly less temperature sensitive in the boreal zone than the other models, in agreement with the results in our study. Flux observation - based global and northern latitude studies by Knox et al. (2019, 2021) and Peltola et al. (2019) also emphasized the importance of temperature in controlling the wetland emissions, though water table level might become important at sites where water table level is below surface for a significant part of the year.

The summer months with the highest mean temperatures were not always anomalous in CTE-CH4 posterior fluxes. JSBACH-H predicted the largest prior emissions during the warmest months. In the posterior and especially in July 2018 the fluxes were decreased, possibly because of the decrease in soil water table level as June and July 2018 suffered from a lack of precipitation (see e.g. Peters et al., 2020). Rinne et al. (2020) also noted that methane emissions in four out of five Fennoscandian wetland sites were decreased in 2018 due to a decrease in water table levels. This suggests that many wetlands suffered from drought, which could have affected the regional results. The summer months with high precipitation often resulted in high posterior

emissions. August was the month of the average seasonal precipitation maximum, while the average temperature maximum was in July. Large increases in the posterior were assigned to high precipitation periods in e.g. August 2008, 2016 and late summer 2007, however the year 2011 with observed high methane emissions from upland soil in northern Fennoscandia

(Lohila et al., 2016) did not stand out in posterior emissions. It is possible that the signal observed by Lohila et al., (2016) was not seen in the larger region of this study, but only in northern Fennoscandia.

The CTE-CH4 inversions quite unanimously attempted to move the seasonal maximum of the emissions towards August. This is supported by the GCP inversion ensemble and the observation-based up-scaled eddy covariance fluxes, mapped over

Fennoscandia. Warwick et al. (2016) also found that seasonal cycles of methane mixing ratios at northern high latitudes above 50° N were improved when the seasonal maximum in northern high-latitude wetland emissions predicted by process models was delayed by one month from July to August. In our work, many models had their seasonal maxima in July or August, notable exceptions being CLM4.5 and CLM5 (bias towards spring) and LPX-Bern (bias towards autumn). The dominance of mineral land over peatland emissions (such as in LPX-Bern) may delay the month of the maximum emissions, as well as using

a large wetland extent in late summer. Placing more peatlands in the southern parts of the region (like in GCP-diag) or having a weak temperature response may possibly result in a longer seasonal emission maximum. A pronounced inundation period after snow melt could induce large methane emissions in spring (like in CLM4.5 and CLM 5). Phenology may also play a role, however modelled start of the growing season was usually delayed from satellite observations in northern latitudes (Peano et al., 2021), which means that the early methane emission maximum is not caused by too early start in model photosynthesis.

According to flux measurements at boreal peatlands, the month of highest emissions was July or August depending on the year (e.g. Rinne et al., 2020). Lake emissions are often not present in process models but are seen by inversions. They might delay the emission maximum in dry years when wetland emissions diminish towards end of the summer due to decreasing soil water table level but lake emissions continue. The CTE-CH4 posterior fluxes, the GCP model ensembles and the observation-based up-scaled eddy covariance fluxes all indicate an emission maximum in August. This provides a coherent view of the seasonality

in contrast to the large variation in ecosystem model results.

## 5. Conclusions

The ecosystem models showed variable responses of methane emissions to temperature and precipitation for the Fennoscandia region. However, multi-model means, inversions and up-scaled eddy covariance flux observations agreed on the month of

maximum emissions and had rather balanced temperature and precipitation responses (i.e. both temperature and precipitation explain the variance of fluxes) which were not significantly changed from prior to posterior in inversions. When two models with contrasting response patterns were used as priors to inversion, the inversion attempted to move emissions of both in posterior towards co-limitation of temperature and precipitation, i.e. to be sensitive to changes in both environmental variables.

The set-up of different emission components (peatland emissions, mineral land fluxes) is important in building up the response patterns, as they contribute to the total flux seen by inversions. Peatland emissions determine the month of maximum emissions in the models that are more sensitive to temperature, while wet mineral soil emissions determine the timing of the maximum in the case of strong precipitation sensitivity. This applies to average response patterns when data from several years is used to determine the responses, noting that the models are sensitive to precipitation in the cases of severe seasonal droughts leading

to water table drawdown in peatlands, which in turn leads to reductions in methane emissions. Depending on the model, wet mineral soil and inundated land emissions can modify the seasonality of methane emissions together with peatland emissions. Peatland emissions are considered to be the main component of methane emissions and wet mineral lands are often not considered to be as important. However, despite the smaller emission per unit area the wet regions emitting methane can be very large, and thus they can change the seasonality of the regional emissions. Furthermore, the wet mineral lands are

implemented in different ways in the models and thus may produce different outcomes for the total flux. Therefore, it is essential to pay more attention to the role of the individual emission components, their magnitude, annual cycle and spatial extent in different regions, and in general consider how the fluxes should be scaled up from site to region (see also Bansal et al., 2023, Knox et al., 2020, Treat et al., 2018, Tuovinen et al., 2019).   To perform well in the Fennoscandia region, it is expected that a model will need to consider peatland and mineral land source and sink components of methane emissions and

use up-to-date land cover description. If the peatland extent is simulated by the model, it needs to be validated against land cover data from bottom-up inventories.  Similarly, for soil wetness or inundation, the model should be validated against satellite data wherever possible. The models should use process-based descriptions for both mineral and peat soil fluxes to simulate the flux responses to climate drivers. Furthermore, it is important to study the overall responses of the total emissions to air temperature and precipitation, as it defines the response of wetland emissions to climate change.


*Data availability*

The data processed for this study will be available on the FMI Research Data Repository METIS (https://fmi.b2share.csc.fi/, DOI: 10.57707/fmi-b2share.23392f65b1354d49abd5146aa5589f9b).

*Author contribution*

TA designed the study, processed and analysed the data, and wrote the paper, AT, JM, MT and VM processed and analysed data, AT, JM, EB, SC, HL, AL, TM, SM, PM, DP, OP, BP, MR, MS, DW and SZ performed model simulations and/or made simulation data available, and AT, JM, JM, MT, VM, YG, TK, HL, AL, TM, SM, PM, DP, OP, BP, MR, MS, DW and SZ reviewed and commented on the manuscript

*Competing interests*

At least one of the (co-)authors is a member of the editorial board of Biogeosciences

*Acknowledgements*

We would like to thank the Academy of Finland projects (307331, 337552, 345531, 351311), European Space Agency ESRIN (Contract No 4000125046/18/I-NB MethEO, 4000137895/22/I-AG MethaneCAMP, AO/1-10901/21/I-DT AMPAC-Net), EU-H2020 and Horizon projects (641816, 776810, 101056844, 101056848 and 101081395) and EU LIFE21-CCM-LV-LIFE

PeatCarbon – 101074396 for financial support. T.K. acknowledges support from the German Federal Ministry of Education and Research (BMBF) through the project Palmod, grant no. 01LP1921A. $CH_4$ In situ observations collected over the US Southern Great Plains were supported by the Office of Biological and Environmental Research of the US Department of Energy under contract no. DE-AC02-05CH11231 as part of the Atmospheric Radiation Measurement (ARM) Program, ARM Aerial Facility (AAF), and Terrestrial Ecosystem Science (TES) Program. We acknowledge the data from The Global Methane

Budget 2000-2017 (2020).

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

**Figures**

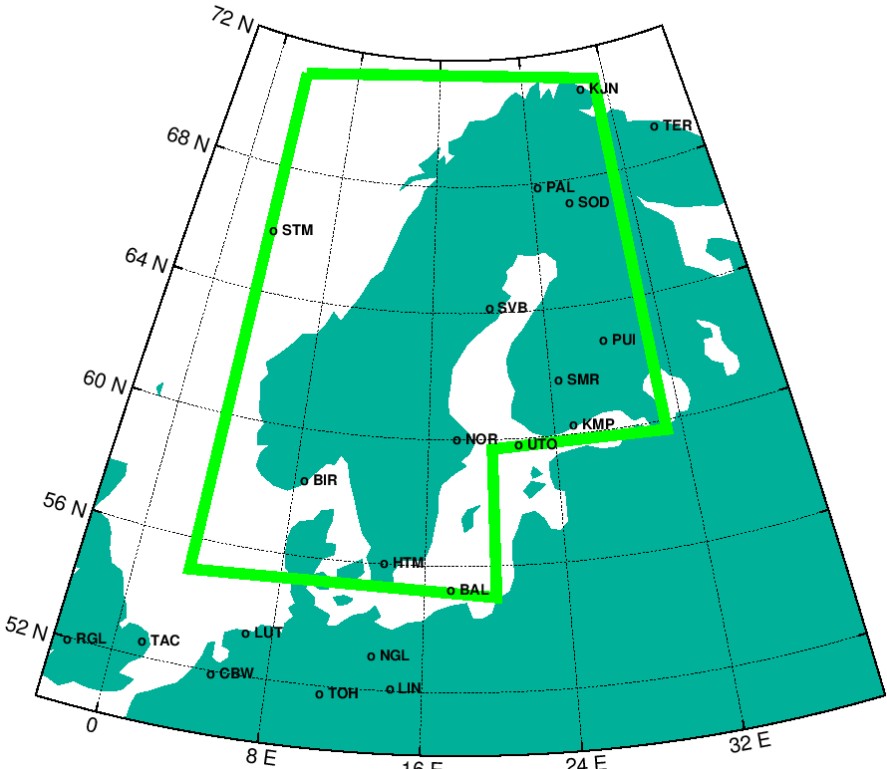

**Figure 1. Study region in Northern Europe.**

a

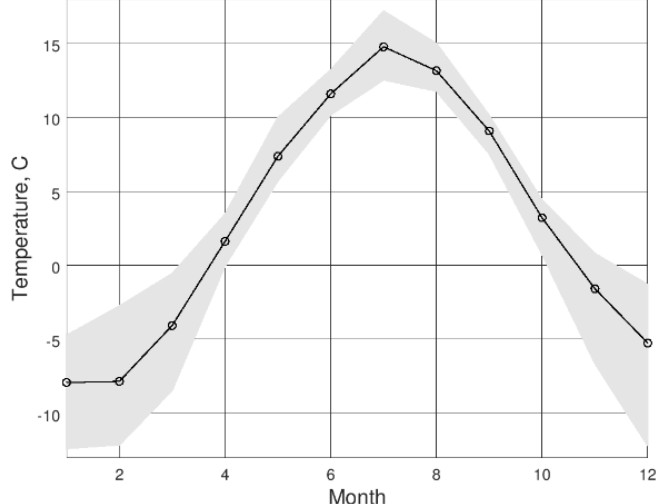


b

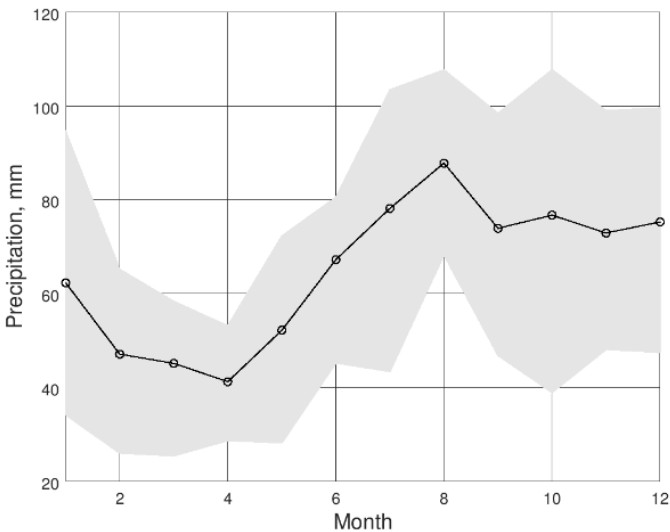

**Figure 2. Mean seasonal cycle of a) temperature and b) precipitation in Fennoscandia over the years 2000-2018 (CRU-JRA dataset).**
**Shading refers to the highest and lowest monthly averages.**

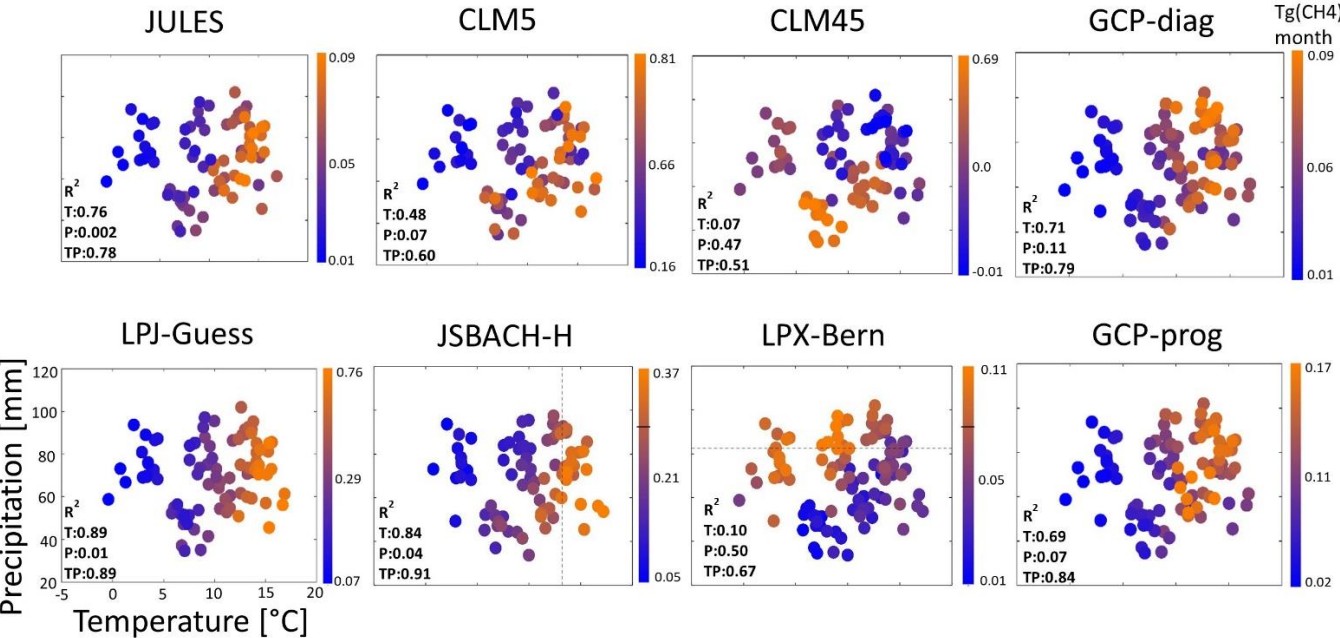

**Figure 3. Temperature and precipitation responses of wetland methane emissions from six ecosystem models and mean of GCP diagnostic and prognostic models in Fennoscandia. Circles refer to monthly averages in May - October during years 2000-2018. R2: Proportion of CH₄ emission variance explained by temperature (T) and precipitation (P) and both together (TP). The dashed line in the LPX-Bern figure shows the 75th percentile of precipitation values. The dashed line in the JSBACH-H figure shows the 75th percentile of temperature values. The black lines in the color scales show the 75th percentile of flux values.**

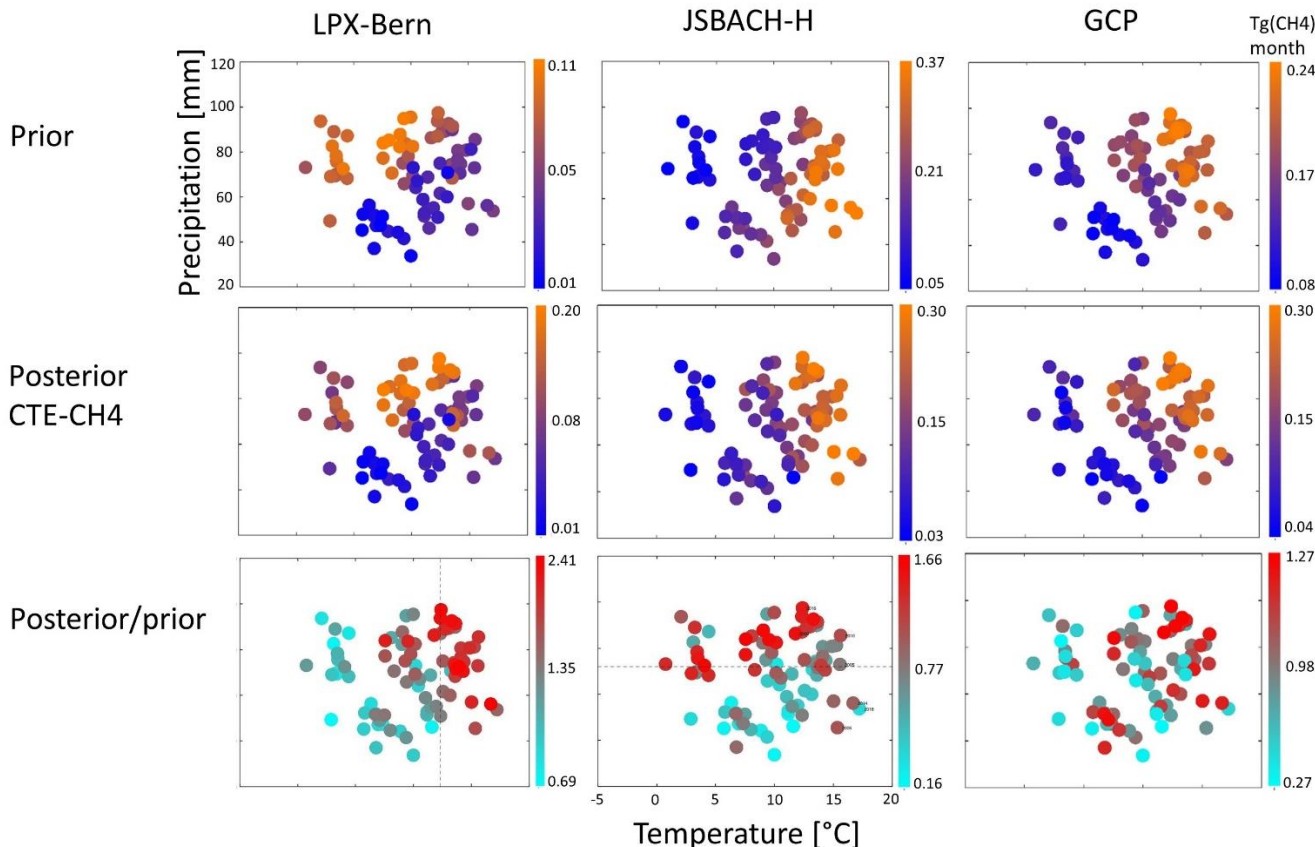

**Figure 4. Temperature and precipitation responses of prior, posterior and posterior/prior methane emissions in Fennoscandia. Circles refer to monthly averages in May - October during years 2000-2018. The line in LPX-Bern posterior/prior figure shows the 64th percentile of temperature values and the highest flux increases from the prior (posterior/prior > 2.0). The line in JSBACH-H posterior/prior figure shows the 51th percentile of precipitation values and the highest flux increases from the prior (posterior/prior > 1.0)**

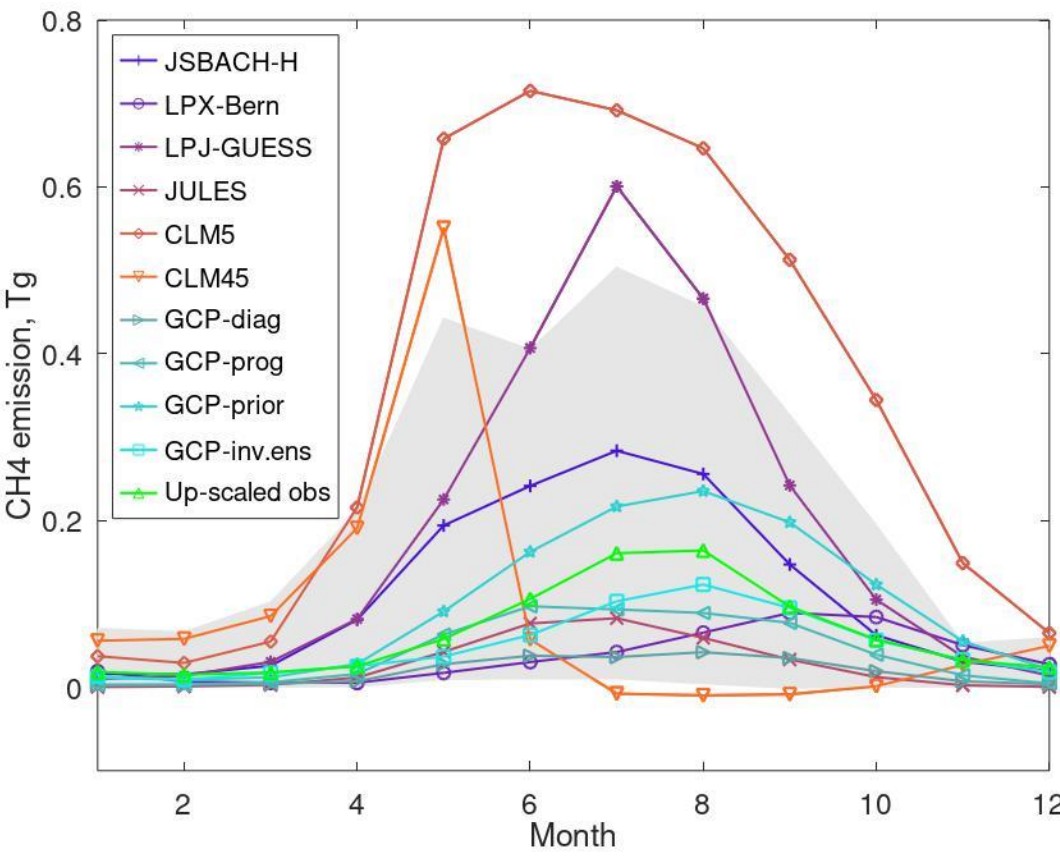


**Figure 5. Seasonal cycle of methane emissions in Fennoscandia according to ecosystem models, mean of GCP diagnostic, prognostic, prior, inversion ensemble models (Saunois et al., 2020) and up-scaled eddy covariance flux observations (Peltola et al., 2019). Shading refers to the largest and smallest members of the GCP diagnostic model ensemble.**



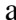

a

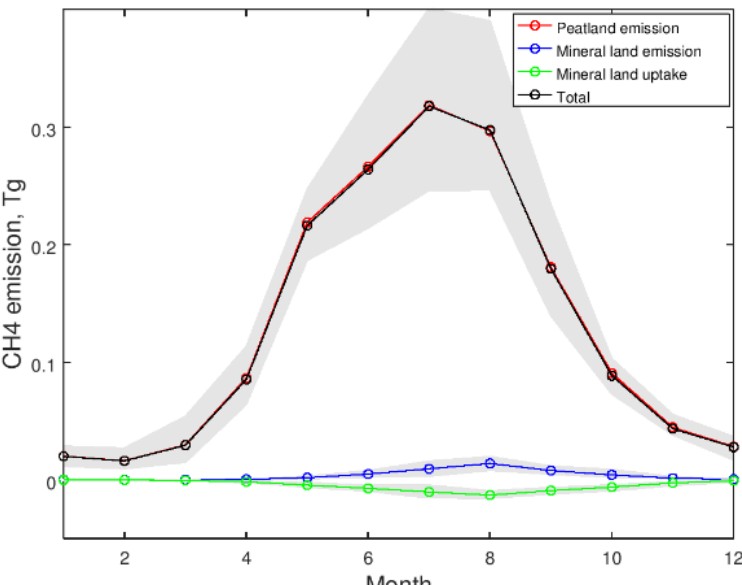

b

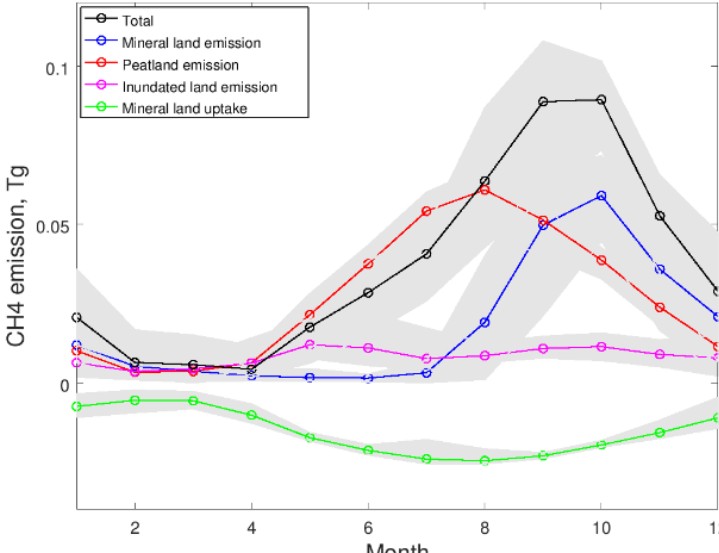

**Figure 6. Seasonal cycle of CH₄ emission components in Fennoscandia for a) JSBACH-H and b) LPX-Bern. Shading refers to maximum and minimum monthly emissions over years 2000 - 2018.**


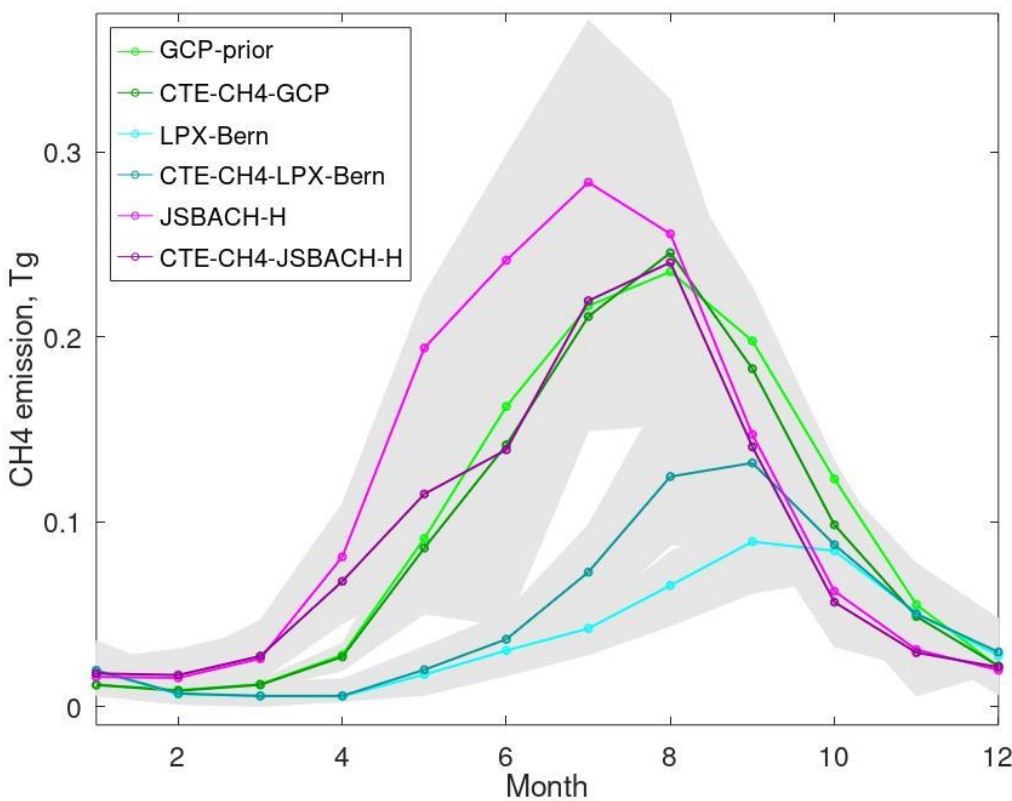

**Figure 7. Seasonal cycle of the wetland CH₄ emissions in Fennoscandia according to the CTE-CH4 inversion model with JSBACH-H, LPX-Bern and GCP-prior as priors. Shading refers to maximum and minimum monthly emissions over years 2000 - 2018.**


**Tables**

| Model | Type | Institution | Time period | Resolution | Reference |
|---|---|---|---|---|---|
| JSBACH-HIMMELI | Ecosystem process model | FMI, Univ. Helsinki, MPI | 2000 - 2018 | 1.875° x 1.875°, 0.1° x 0.1° | Kleinen et al. (2020), Raivonen et al. (2017) |

| LPX-Bern | Ecosystem process model | Univ. Bern | 2000 - 2018 | 0.5° x 0.5° | Lienert and Joos, (2018) |
|---|---|---|---|---|---|
| LPJ-GUESS | Ecosystem process model | Lund Univ. | 2000 - 2014 | 0.5° x 0.5° | Smith et al. (2014) |
| CLM4.5 | Ecosystem process model | NORCE | 2000 - 2014 | 1.25° x 0.9375° | Oleson et al. (2013) |
| CLM5 | Ecosystem process model | CMCC | 2000 - 2014 | 0.5° x 0.5° | Lawrence et al. (2019), Peano et al. (2021) |
| JULES | Ecosystem process model | UKMO, Univ. Exeter | 2000 - 2014 | 0.5° x 0.5° | Gedney et al. (2004), Comyn-Platt et al. (2018), Chadburn et al. (2020) |
| Carbon Tracker Europe – CH4 | Atmospheric inverse model | FMI | 2000 - 2018 | 1.0° x 1.0° | Tsuruta et al. (2017), Tenkanen et al. (2021) |
| Global Carbon Project (GCP) models: - GCP-diag - GCP-prog - GCP-prior -GCP-inversions | Means of - 12 diagnostic ecosystem models, - 8 prognostic ecosystem models, - climatological mean inversion prior - 5 atmospheric inverse models | GCP | 2000 - 2017 | Fluxes processed to 1.0° x 1.0° | Saunois et al. (2020), Poulter et al. (2017) |

| Up-scaled fluxes | Machine learning based random forest model | FMI | 2013 - 2014 | 1.0° x 1.0° | Peltola et al. (2019). |
|---|---|---|---|---|---|


**Table 1: Models and set-ups**
