# Peer review of "Air temperature and precipitation constraining the modelled wetland methane emissions in a boreal region in Northern Europe"

_EGUsphere, 2023_

## Author Comment (AC1)

*We thank the reviewers for the detailed and constructive comments. We agree with most of the suggestions and will make corresponding changes. We hope that our answers are sufficient and believe the manuscript will be much improved after the corrections. Detailed answers can be found in the files attached.*

Reviewer 2

The authors investigated how simulated CH4 emissions from six ecosystem models respond to temperature and precipitation in Northern European regions. Then, simulated CH4 emissions from two ecosystem models exhibiting contrasting response patterns, and the ensemble mean of GCP atmospheric inversions, were used as priors of wetland emissions in an inversion. By doing this, they explored how the inversion changes the fluxes and alters the response of CH4 emissions to temperature and precipitation.

General comments:

Climate forcing used by the six ecosystem models are different. Previous studies have shown significant variations in process-based LSM outputs stemming from the choice of climate forcing. Four out of the six models were driven by CRUNCEPv7, one was driven by CRU-JRA and another one was driven by CRU-HARMONIE.

If my understanding is correct, temperature and precipitation data from CRU-JRA were applied to all six ecosystem models and the mean of GCP models, when studying the responses of simulated CH4 emissions to temperature and precipitation (Fig. 1). Why not use the specific climate inputs that drive each model?

*To address this, we studied the effect of the climate input using results from uncoupled and coupled models (manuscript Fig. S1), where the driver data was either CRU-JRA for the uncoupled case or the ESM-specific climate model input for the coupled case. It would be expected that using coupled climate data as model input would lead to greater differences than simply using different uncoupled data sets, as all the uncoupled datasets (CRUNCEPv7, CRU-JRA and CRU-HARMONIE) are bias-corrected using precipitation and temperature measurements. The ESM-specific climate model data is not bias-corrected against such data. However, the response patterns were not significantly different (p<0.01). We are therefore confident that using CRU-JRA consistently throughout the paper produces robust results. To further test the impact of two different bias-corrected data sets, we created a figure (Fig. 1), where CRU-HARMONIE and CRU-JRA are plotted against JSBACH results, showing only small differences regarding the response of methane emissions to temperature and precipitation.*

[Figure]

*Figure X1: JSBACH ecosystem model results, plotted against left: CRU-HARMONIE ($R^2\_Temp$ =0.822, $R^2\_Prec$ = 0.054, $R^2\_TP$ = 0.834), right: CRU-JRA climate data ($R^2\_Temp$ =0.823, $R^2\_Prec$ = 0.0045, $R^2\_TP$ = 0.829).*

The representation of wetland area and inundation were various among models. Some models used a static and prescribed wetland area, some models used prescribed but time-varying wetland area, and other models dynamically simulated inundation and wetland area. It would be very useful to have a figure to show the wetland area used by each model.

*We will add figures of wetland areas of the models. As the area of inundated land and wet mineral land varies over time, we will provide example maps for the most relevant time periods. The combined wetland area map including peatlands, inundated lands and wet mineral lands is quite different compared to the methane emission maps, because wet mineral land can cover wide land areas but their emissions per unit area are smaller than those from peatlands (See example figures X2 and X3 below for JSBACH-HIMMELI model). Mineral soil was considered to be wet when the daily soil moisture, simulated for the 0.1 x 0.1 degree resolution grid cell, was high and there were methane emissions. Then the wetland fraction of the land area in that grid cell was 1.0, as the land area comprised of wet mineral land, inundated land and peatland, all emitting methane. The wet fraction of mineral lands was calculated from the individual daily 0.1 x 0.1 degree resolution grid cells up-scaled and averaged over the month in question and over years 2000 - 2018.*

[Figure]

*Figure X2: Methane emitting land areas in JSBACH-HIMMELI model, including a) peatlands, b) peatlands + inundated lands and c) peatlands + inundated lands + wet mineral lands. The wetland fraction is calculated as an average for the month of August over years 2000 – 2018.*

[Figure]

*Figure X3: Methane emitting land areas (including peatlands + inundated lands + wet mineral lands) in JSBACH-HIMMELI model, for the month of a) July, b) August and c) September. The wetland fraction is calculated as an average for the month in question over years 2000 – 2018.*

While the authors acknowledged the importance of wetland area in determining boreal regions CH4 emissions in both the introduction and the discussion section, the comparison of models' outputs and the analysis of precipitation and temperature responses didn't address the impact of wetland area.

*We will discuss the wetland area in the results section; it was embedded in the discussion of emissions from wet mineral soils and inundated land, as the magnitude of those is determined by the extent of the wetland area.*

*See e.g. section 'Model components and seasonal cycle' for LPX-Bern: 'The soil moisture and consequently the wet mineral land area peaked in autumn, and thus the wet mineral land emissions were at maximum in October in contrast to peatland emissions,….*

Figure 7 shows substantial difference among models in terms of both the magnitude and the seasonal cycle of CH4 emissions. The influence of climate forcings and wetland area on these differences should be explicitly discussed.

*We will add the following text on the impact of the wetland area, climate forcings and model features on the seasonal cycles:*

*Note that each model presents differences in seasonal cycle and peak month as they differ in wetland area (Figure: wetland area) and model features. On the contrary, the use of different climate-forcing data products or results from coupled climate simulations show minor differences in methane emission responses (Figure X1and existing Figure S1).*

Though the authors showed both prior and posterior wetland CH4 emissions from the inversion model in Figure 6, as a reader not familiar with inversion models, I still find it challenging to grasp how the prior wetland CH4 emissions are adjusted. To achieve changes in both the magnitude and seasonal cycle of prior wetland CH4 emissions as shown in Figure 6, what other sources or sinks of CH4 have been altered?

*The inversion model used in the current work adjusts the emissions of CH4 every week. Thus, in addition to optimizing the magnitude of the emissions, the seasonal cycle is also adjusted. In addition to biospheric emissions inversions also optimize the anthropogenic emissions, but these have much less seasonal variability. Anthropogenic emissions are optimized separately but at the same time as natural emissions so that the total emissions produce modelled atmospheric concentrations that are consistent with concentration observations. There are also minor sources that are not optimized, from e.g. fires, termites and ocean, and the atmospheric sink which is calculated using prescribed data obtained from a chemistry model (e.g. OH fields).These details will be added to the description of CTE-CH4.*

Specific comments:

L266: explained CH4 emission -> CH4 emission

*We will correct the wording*

L338: Fig 1 should be Fig 2

*We will correct the number*

L392: to -> of?

*We will correct the wording*

Figure 3: A figure for upscaled flux observations could be added to show observation-based temperature and precipitation responses of wetland CH4 emissions.

*A figure will be added (Fig. X4 below), using data from Olli Peltola et al 2019. There are not as many summer months of data as for the models, but the results suggest that the fluxes are correlated with both temperature and precipitation ($R^2\_Temp$ =0.83, $R^2\_Prec$ =  0.54, $R^2\_TP$ = 0.91).*

.

[Figure]

*Figure X4. Temperature and precipitation responses of upscaled methane emissions in Fennoscandia. Circles refer to monthly averages in May - September during years 2013-2014.*

Figure 6: The seasonal cycle of in situ atmospheric CH4 observations could be included in this figure to aid in understanding why the CTE-CH4 inversions shifted the monthly flux maximum towards August.

*The seasonal cycle of CH4 observations also has components other than biospheric emissions; anthropogenic emissions and atmospheric OH chemistry. The OH chemistry sink has a strong seasonal cycle with maximum impact in spring (see Fig X5 below)  and therefore the month of the summer maximum of the biospheric emissions is sometimes not clearly visible, and the visibility also depends on where the air masses have been transported. The summer maximum can often be seen as elevated concentrations in late summer, however it is minor compared to the winter maximum (which is caused by the combined effect of anthropogenic emissions continuing throughout the year, and a weak wintertime OH sink).   Therefore, adding a set of observations from different sites would not help much, but would make the figure more unclear for the reader. The inversion model solves for concentration changes due to anthropogenic and biospheric emissions, OH sink and atmospheric transport.*

[Figure]

*Figure X5. Methane concentrations measured at Pallas Sammaltunturi station in Finland.*

Figure 7: It is difficult to distinguish lines for LPX-Bern and LPJ-GUESS, lines for the four GCPs, due to similar colors.

*We will modify the lines for LPX-Bern and LPJ-GUESS and GCPs to be clearer*

---

## Author Comment (AC2)

*We thank the reviewers for the detailed and constructive comments. We agree with most of the suggestions and will make corresponding changes. We hope that our answers are sufficient and believe the manuscript will be much improved after the corrections. Detailed answers can be found in the files attached.*

**Reviewer 1**

The authors present an assessment of the impact of temperature and precipitations on methane emissions by Northern European wetlands as simulated by 6 process models. This study makes use of a set of ecosystem models, atmospheric inversions and fluxes deduced from eddy-covariance measurements to gain insights on the behavior of the 6 models with regard to temperature and precipitations.

**General comments**

The protocol of the study, with the various inter-comparisons, is very good, making use of all possible/relevant ways to assess the behavior of the 6 targeted models. It will be very useful to a wide community, working on process models but also on other models and even atmospheric inversions.

Nevertheless, because it should be usable by people with different backgrounds and probably also due to the many co-authors from different approaches, the structure of the manuscript, the order in which the ideas and information is delivered must be revised. Inside many sections and subsections, there are missing links between the ideas or pieces of information. This is not helped by the use of verbs in the past tense which should probably be revised by a native speaker. Even though I am not a native speaker, I was confused by this point about which information is from what was done before the study or not in the scope of the paper, what has been done for the study during the preparatory phase and what has been learned from the study.

*Here we kindly note that the co-authors include three native speakers who revised the manuscript and corrected the language.*

**Specific comments**

**Abstract**

I think the abstract may be rewritten to better represent the quality and important results of the study: some key sentences are too vague or general, as detailed below.

- the whole abstract: why not write in the present tense?

*We will change to present tense*

- l.31: "compared to multi-model mean": a multimodel mean = one reference?

*There is more than one multi-model mean, we will change to multi-model means*

- l.34-35: "how the inversion attempts to change the prior fluxes in the posterior": clumsily stated, not very clear for readers outside inversions. Please rephrase, possibly on the lines of how the assimilation of atmospheric data correct the fluxes.

*We will change to:*

*how the assimilation of atmospheric concentration data changes the flux estimates*

- l.36: "to move emissions": i.e. some of the emissions or all emissions?

*We will change to:*

*wetland emissions*

- l.36: "in posterior": this is probably too specific a word for an abstract of a paper not aimed at the inversion community. Please rephrase.

*We will remove the word*

- l.36-37: "in general", "often": this is too vague, please quantify or at least, indicate in space, in time...

*See below*

- l.37-38: "This was not the case for the warm and dry period of summer 2018": with in general and often in the previous sentence, this sentence does not bring any information.

*We will change to:*

*Between 2000 and 2018, periods of high temperature and/or high precipitation often resulted in increased emissions. However, the dry summer of 2018 did not result in increased emissions despite the high temperatures.*

- l.39-40: "varied from May to September": from one year to another? From one model to another? Please clarify.

*We will change to:*

*The month with the highest emissions varies from May to September among the models.*

- l.41:"balanced": what is a balanced response? Please define or rephrase.

*We will change to:*

*However, multi-model means, inversions and up-scaled eddy covariance flux observations* *agree on the month of maximum emissions and are co-limited by temperature and precipitation*

- l.42,43: "significant": define/quantify what is significant in the context. Or avoid this word outside a clear statistics framework.

*We will change to: important*

- l.43-44: "it is essential to pay more attention to the magnitude, composition, annual cycle and climate driver responses of wetland emissions in different regions." I guess everybody working on process-models knows about this. Isn't there a more practical or precise conclusion from this work?

*We will revise the wording of the sentence to be more precise:* *Considering the significant differences among the models, it is essential to pay more attention to the regional representation of wet and dry mineral soils and periodic flooding which contribute to the seasonality and magnitude of methane fluxes. The realistic representation of temperature dependence of the peat soil fluxes is also important. Furthermore, it is important to use process-based descriptions for both mineral and peat soil fluxes to simulate the flux responses to climate drivers.*

**Introduction**

As such, the introduction does not make it very clear how models use input information and what the issues are with temperature and precipitations.

'Clarification on how models use input information':

*L46 We will change to:*

*Temperature, soil moisture, water table depth and primary production drive the carbon accumulation, respiration and methane emissions from peatlands* *and are modelled by ecosystem process models using atmospheric climate data such as temperature and precipitation as input to the simulations*

'Clarification on issue with temperature and precipitation':

*We assume the issue refers to clarifying the temperature and precipitation responses in a regional context, as this article is not addressing the calibration of individual models at specific sites and therefore is not dealing with any specific issue with calibration of temperature or precipitation responses, just diagnosing the characteristics of the models when applied to estimation of regional CH4 fluxes. We address the importance of regional temperature and precipitation studies in the following:*

*L68-73…The model predictions of regional annual cycles of methane emission differ significantly and the future estimates of the total global methane emissions are highly variable (Stocker et al., 2013, Saunois et al., 2020). Therefore it is useful to take a more climate-oriented perspective to the drivers of the methane emission in order to make better predictions of the responses to future climate change (Koffi et al., 2020), and to emphasize the regional approaches. It is important to study the responses of the emissions to air temperature and precipitation, as it defines the response of wetland emissions to climate change.*

Regarding the order in which the pieces of information are delivered, first, a presentation of the various types of wetlands in the real world would be useful - it is done partly in the Results, when listing what types are taken into account in some models; then, a presentation of what is done in the models (for example, ignore lakes).

*Due to complexity of wetlands, there are various definitions for different types of wetlands, and they vary from one region or country to another. Harmonization of the definitions and creation of a consistent wetland map is a topic of more than one on-going projects. Furthermore, the global ecosystem models are not presenting the different types in detail. Therefore we think presenting wetland types in broad categories, which are simulated by the models, are sufficient for this study. We present in the introduction (first paragraph) the different types of wetlands in our models; water-logged peatlands, wet mineral lands and inundated lands.*

*To further clarify, we will add (l46):*

*Wetlands considered in this study include those at peatlands and mineral lands as well as periodically inundated, i.e. flooded lands.*

Overall, is not made clear enough what happens in the real world and what happens in the models, so that it's difficult to understand the challenges this study deals with. Examples: it is difficult to understand why the emissions of process models have not already been validated regarding temperature and precipitations dependencies;

*The process models are often validated against local measurement station data and different models are validated against different local data sets. Here we take a regional approach which is sought for in recent literature and is necessary for global studies, and run several models for the same region, upscaling local fluxes to a regional flux. The temperature and precipitation dependencies may differ in the up-scaled regional fluxes because of differences in model set-ups, calibration and validation, and presence of the different wetland types and distributions in the region. We address the challenges related to regional fluxes in the introduction and references therein (see also above ' issues with temperature and precipitation').*

the way inversions can inform process models is presented in a misleading way: too optimistic if it's flux inversion, not clear if it's process models' parameters which are inverted.

*Here we are looking at the regional up-scaled fluxes and responses. We do not claim that inversions can inform process models on their responses at site level or optimize their parameters. It is also true that the results from atmospheric inversion is not a ground truth. However, for regional fluxes atmospheric inversions provide more consistent results with atmospheric state (concentrations), and therefore, could be taken as one of very few means for evaluation of process*

*models. Although atmospheric inversion does not provide direct information about how process models' parameters or climate responses should be improved, we argue that the evaluation with inverse models could give a guidance towards which the process models should be improved.*

*We will add:*

*Atmospheric inverse models relying on atmospheric methane concentrations provide a top-down regional view of the responses of methane emissions to climate drivers, attempting to detach them from the underlying prior assumptions. In process models the responses are more subject to how the processes were built and dependencies constructed, and how the fluxes were up-scaled. Therefore, the evaluation with atmospheric inversion models could provide useful information to process models on to which direction they should improve their regional emission estimates and climate responses.*

- l.46: "the second most important greenhouse gas": ANTHROPOGENIC!

*We will remove the sentence*

- l.48: "peatlands": please list the types of wetlands before going into details for each of them.

*We will add:*

*Wetlands considered in this study include those at peatlands and mineral lands as well as periodically inundated, i.e. flooded lands.*

- l.52: "There are accurate peat-land maps": what about the mineral lands, mentioned just before but never again discussed?

*We will change the order of sentences in the paragraph to make the discussion of mineral lands more connected, and add sentences:*

*There are accurate peatland maps for the northern regions based on in situ data of peat layer thickness (e.g. Xu et al., 2018, Tanneberger et al., 2017), which enable estimations of the peatland methane emissions by process models if the soil water table level and soil carbon processes providing substrate for methane production are well represented. Land not covered by peatlands includes mineral land. In addition, mineral lands can act as a source of methane if the soil is very moist or inundated (Lohila et al., 2016, Wolf et al., 2011, see also Bansal et al., 2023), with a significant contribution from the organic layer on top of the soil. The soil moisture and land inundation can also be estimated by models together with peat accumulation, though it is still challenging (e.g. Loisel, et al., 2021, Ito et al., 2020). Soil moisture is an important input variable for mineral soil emission modelling (e.g. Curry et al., 2007).*

- l.57: "this feature is badly represented": this seems contradictory with the "accurate peat-land maps" of the previous paragraph. Explain more clearly what is know accurately and what is poorly known.

*Accurate peat-land maps are only one part of methane emission estimations; the soil hydrology (water table level) also needs to be simulated, and that is poorly represented. The extent of inundated lands is badly represented in the boreal zone because of forest canopy blocking the satellite view and lakes and ponds being interpreted as inundated land, as mentioned later in the text*

*We will add:*

*There are accurate peatland maps for the northern regions based on in situ data of peat layer thickness (e.g. Xu et al., 2018, Tanneberger et al., 2017), which enable estimations of the peatland methane emissions by process models, if the soil water table level and soil carbon processes providing substrate for methane production are well represented.*

- l.67: "significantly": by how much?

*We will remove the word*

- l.68: "more climate-oriented": i.e. instead of using maps of the extent? Please explain more clearly.

*We will change to:*

*Therefore, instead of studying response to wetland extent, it is useful to take a more climate-oriented perspective to the drivers of the methane emission*

- l.69: "to emphasize the regional approaches.": what does it mean?

*We will modify the sentence:*

*Therefore, instead of studying response to wetland extent, it is useful to take a more climate-oriented perspective to the drivers of the methane emission in order to make better predictions of the responses to future climate change (Koffi et al., 2020). Further, it is important to emphasize the regional approaches as the drivers of emissions vary widely in their spatial distribution, climate and ecosystem type (Stavert et al., 2020)*

- l.70: "to study the responses of the emissions to air temperature and precipitation": the process models are built to take temperature and precipitation into account, aren't they? Haven't they be validated on these points? Make the challenge(s) clearer!

*As noted earlier, they are not validated by their regional responses. We will clarify the text:*

*It is important to study the responses of the regional emissions to air temperature and precipitation, as it defines the response of regional wetland emissions to climate change.*

- l.74: "arises": is this the right word?

*We will change to:*

becomes apparent

- l.75: "inundation": please define.

*We will change to:*

periodical inundation, i.e. flooding

- l.81: "according to atmospheric inversion modeling": this is too vague. Atmospheric inversions as such are not able to point at the causes/drivers of the corrections they apply to fluxes. So probably something more has been done than raw inversions by Thompson et al.

*We will add:*

and analysis of the results using climate reanalysis data (Dee et al., 2011).

- l.85-86: "provide a top-down view of the responses of methane emissions to climate drivers, attempting to detach them from the underlying prior assumptions.": not clear, even for somebody working on flux inversion. What is inverted here? Methane fluxes? If so, insights on climate drivers are not easy to obtain and the prior assumptions play a large part!

*We will clarify the text by modifying sentences:*

*Atmospheric inverse models rely on atmospheric methane concentrations, and they provide a top-down view of the methane emissions. Their results can be used to study responses of regional methane emissions to climate drivers.*

- l.87-88: "atmospheric inversion models can be used to inform process models on how they should improve their emission estimates and climate responses": not so simple, if it's flux inversions: they can indicate where fluxes should be larger or smaller than what the process models compute but neither why they are too small or too large nor how to correct the process models. This last point is provided by inversion not of fluxes but of parameters of the process models. Please be very precise on the inversions you are dealing with.

*It is true that atmospheric inversion models do not directly inform why fluxes from process models are too small/large (e.g. whether it is due to drivers, bad parameter values or missing some processes), nor how to correct the parameters. Here, we meant to argue that the atmospheric inversion model results could also be studied with e.g. climate or ecosystem data and explain the relationships. Then the results about the relationships between fluxes and climate or ecosystem data can be compared to those from process models. Such comparison is worthwhile for informing process model to which direction they should improve their models. The inversion models themselves are not mechanistically explaining anything, but their flux results, based on assimilated atmospheric data, include the information needed to train a response model.*

*We will change to:*

*Therefore, it is worthwhile comparing the atmospheric inversion models to process models and study together how their regional emission estimates and climate responses differ.*

- l.89: "to study the responses of the emissions to air temperature and precipitation": I would think this is part of the evaluation/validation of process models! Please make the challenges clearer.

*We will removed the sentence, as it somewhat repeated what was already said earlier.*

- l.91-92: "the ensemble of models from the Global Carbon Project (GCP) 2020 estimation": of which the Crescendo models are not part?

*See below (Materials and methods reply 1 marked with a star)*

- l.93-96: please cut up this sentence into several manageable shorter ones.

*We will divide the sentence into shorter ones.*

**Materials and methods**

At least one model is in the ensemble used for comparison: please explain how this is not an issue for this study.

*\*The Global Carbon Project model ensemble provides a collection of best estimates from individual models. The models may share similar components so they are not fully independent. The varying model set-ups may also produce different results from an earlier use case. Their predictions are not random but the probability to having systematic biases is smaller in comparison to a single model. The uncertainty of the model ensemble is considered to be smaller, or at least better known than the uncertainty of a single model, and as the size of a model ensemble increases the reliability of the result (may) also increase. To increase the size of an ensemble it is in our case necessary to include all models that have participated in GCP, also if they are variants of the same model like in the case*

*of JSBACH, partly including same model components. The model estimates from Global Carbon project are used as a mean of several models, and they create a smoothed version of the temperature and precipitation responses. The individual model response patterns are reflected to the mean and finding the deviations is an important aspect of this work. Therefore we find it acceptable to use the full GCP model ensemble mean.*

At the end of each description, or better still, in an overall summary (maybe in a table), it would be very useful to have the limitations (e.g. neglected processes, poorly known inputs) and advantages of each model compared to the others (not need to list what they all do the same way) with regard to methane emissions. Maybe it would even be possible to state what is expected from each model e.g. performs better in a given region or season. I'd expect the impact of the spatial resolution to be large - but I don't run process models.

*We are comparing the models by their results in Discussion (line 360 ff): 'JSBACH-H, LPJ-GUESS and JULES were clearly more constrained by temperature. The reason behind this behaviour could be linked to strong temperature dependencies in the process descriptions (production, oxidation, transport)...'*

*We would like to be cautious in listing the limitations/advantages of the models in a deep process level and stating that some are better or worse than others, as we are studying regional temperature and precipitation responses which is only one aspect of model performance. For a full evaluation we would need to make a specifically designed benchmarking study. For example, we do not have the possibility to study the effect of spatial resolution, as we did not perform experiments with the same model set-up running in different resolutions. Additionally, this study partly uses published data, such as that from GCP, where the global limitations and advantages are discussed in detail. Furthermore, each model already has their own detailed description and calibration papers.*

*We will extend the conclusions to give some general expectations for a model to perform well in the study region.*

Please make the paragraphs of the various models homogeneous, with the same structure e.g. first the general history of development, then the relevant points for methane emissions and in the end, the sets of emissions used on this study with the name used in the remainder of the text.

*We will rearrange the sentences in model descriptions from LPJ-GUESS and JULES part*

Subsections 2.1 to 2.7 : it is not clear how many runs or set-ups from each model are used in the study, please provide an overall summary, maybe as a table, with consistent tags. An assessment of the differences between the runs/set-ups and how they are expected to impact methane emissions and their response to temperature and precipitation is expected by the reader.

In several sections, the order of presentation is misleading i.e. the inputs are described after (part of) the description of the results. Please re-order.

*We will add a table (Table X1 below) describing the models and set-ups:*

*Table X1: Models and set-ups*

| Model | Type | Institution | Time period | Resolution | Reference |
|---|---|---|---|---|---|
| JSBACH-HIMMELI | Ecosystem process model | FMI, Univ. Helsinki, MPI | 2000 - 2018 | 1.875° x 1.875°, 0.1° x 0.1° | Kleinen et al. (2020), Raivonen et al. (2017) |
| LPX-Bern | Ecosystem process model | Univ. Bern | 2000 - 2018 | 0.5° x 0.5° | Lienert and Joos, (2018) |
| LPJ-GUESS | Ecosystem process model | Lund Univ. | 2000 - 2014 | 0.5° x 0.5° | Smith et al. (2014) |
| CLM4.5 | Ecosystem process model | NORCE | 2000 - 2014 | 1.25° x 0.9375° | Oleson et al. (2013) |
| CLM5 | Ecosystem process model | CMCC | 2000 - 2014 | 0.5° x 0.5° | Lawrence et al. (2019), Peano et al. (2021) |
| JULES | Ecosystem process model | UKMO, Univ. Exeter | 2000 - 2014 | 0.5° x 0.5° | Gedney et al. (2004), Comyn-Platt et al. (2018), Chadburn et al. (2020) |
| Carbon Tracker Europe – CH4 | Atmospheric inverse model | FMI | 2000 - 2018 | 1.0° x 1.0° | Tsuruta et al. (2017), Tenkanen et al. (2021) |
| Global Carbon Project (GCP) models: - GCP-diag - GCP-prog - GCP-prior - GCP-inversions | Means of - 12 diagnostic ecosystem models, - 8 prognostic ecosystem models, - climatological mean inversion prior - 5 atmospheric inverse models | GCP | 2000 - 2017 | Fluxes processed to 1.0° x 1.0° | Saunois et al. (2020), Poulter et al. (2017) |
| Up-scaled fluxes | Machine learning based random forest model | FMI | 2013 - 2014 | 1.0° x 1.0° | Peltola et al. (2019). |

Figure 1: provide information on stations, at least their full name and network.

*We will add a table (Table X2 below) providing information on the stations:*

*Table X2. List of surface observation sites used in inversions, located in Figure 1.*

*Data type is categorized into two measurements (discrete (D) and continuous (C)). Date max is limited to 2018/12, which was the last month in inversions.*

| Sitecode | Site Name | Country | Contributor | Longitude | Latitude | Height | Data type | Date min [year/month] | Date max [year/month] |
|---|---|---|---|---|---|---|---|---|---|
| BAL | Baltic Sea | Poland | NOAA/ESRL | 17.22 | 55.35 | 28 | D | 1998/01 | 2011/06 |
| BIR | Birkenes | Norway | NILU | 8.25 | 58.39 | 219 | C | 2009/05 | 2018/12 |
| HTM | Hyltemossa | Sweden | ICOS-ATC, LUND-CEC | 13.42 | 56.10 | 265 | C | 2017/04 | 2018/12 |
| KJN | Kjolnes | Norway | Univ. Exeter | 29.23 | 70.85 | 20 | C | 2013/10 | 2018/12 |
| KMP | Kumpula | Finland | FMI | 24.96 | 60.20 | 53 | C | 2010/01 | 2018/12 |
| NOR | Norunda | Sweden | ICOS-ATC, LUND-CEC | 17.48 | 60.09 | 146 | C | 2017/04 | 2018/12 |
| PAL | Pallas-Sammaltunturi, GAW station | Finland | NOAA/ESRL | 24.12 | 67.97 | 570 | D | 2001/12 | 2018/12 |
| PAL | Pallas-Sammaltunturi, GAW station | Finland | FMI | 24.12 | 67.97 | 570 | C | 2004/02 | 2017/12 |
| PAL | Pallas-Sammaltunturi, GAW station | Finland | ICOS-ATC, FMI | 24.12 | 67.97 | 577 | C | 2017/09 | 2018/12 |
| PUI | Puijo | Finland | FMI | 27.66 | 62.91 | 84 | C | 2011/11 | 2018/12 |
| SMR | Hyytiala | Finland | ICOS-ATC, UHELS | 24.29 | 61.85 | 306 | C | 2016/12 | 2018/12 |
| SOD | Sodankylä | Finland | FMI | 26.64 | 67.36 | 227 | C | 2012/01 | 2018/12 |
| STM | Ocean Station "M" | Norway | NOAA/ESRL | 2.00 | 66.00 | 5 | D | 1999/01 | 2009/11 |
| SVB | Svartberget | Sweden | ICOS-ATC, SLU | 19.78 | 64.26 | 385 | C | 2017/06 | 2018/12 |
| UTO | Uto | Finland | ICOS-ATC, FMI | 21.37 | 59.78 | 65 | C | 2018/03 | 2018/12 |
| Stations outside study region | | | | | | | | | |
| CBW | Cabauw | Netherlands | University of Groningen | 4.93 | 51.97 | 199 | C | 2005/01 | 2013/06 |
| LIN | Lindenberg | Germany | ICOS-ATC, HPB | 14.12 | 52.17 | 171 | c | 2015/10 | 2018/12 |
| LUT | Lutjewad | Netherlands | ICOS-ATC, RUG | 6.35 | 53.40 | 61 | C | 2018/08 | 2018/12 |
| NGL | Neuglobsow | Germany | UBA-Germany | 13.03 | 53.17 | 68.4 | C | 1999/01 | 2013/12 |
| RGL | Ridge Hill | United Kingdom | UNIVBRIS - | 2.54 | 52.00 | 294 | C | 2012/02 | 2017/12 |
| TAC | Tacolneston | United Kingdom | NOAA/ESRL | 1.14 | 52.52 | 236 | D | 2014/06 | 2016/01 |
| TAC | Tacolneston | United Kingdom | UNIVBRIS | 1.14 | 52.52 | 241 | C | 2013/01 | 2017/12 |
| TER | Teriberka | Russian Federation | MGO | 35.10 | 69.20 | 42 | D | 1999/01 | 2018/12 |
| TOH | Torfhaus | Germany | ICOS-ATC, HPB | 10.54 | 51.81 | 948 | C | 2017/12 | 2018/12 |

- l.102: "utilised": is this the right word?

*We will change to used*

- l.104: "further developed in the recent H2020-CRESCENDO project": is there a reference for this e.g. a deliverable and/or report of the project?

*Project reports can be downloaded from https://cordis.europa.eu/project/id/641816/results*

- l.107: "in Global Carbon Project": the ensemble to use as a reference includes at least one of the 6 models targeted here: isn't is an issue?

*Please see the first reply (marked with a star) under Materials and methods*

- l.130: "was set to": based on what data/information?

*For improved peat accumulation rates (will be added in the text)*

- l.133: "manuscript": what does this mean? Currently a draft? Submitted?

*Tyystjärvi et al is in discussion in Biogeosciences (https://egusphere.copernicus.org/preprints/2024/egusphere-2023-3037/). Li et al has been submitted to Tot. Sci. Env. and is currently being revised after the first round of comments. We will modify the status of the manuscripts in the text accordingly.*

- l.145: "suitability for peatland growth conditions": what does this mean?

*We will change to:* **a model that determines peatland growth conditions to simulate** *the peatland spatial distribution*

- l.158: "in general": do you mean it's not the case here? In some set-ups? Or not on some parts of the domain?

*We will remove and rearrange sentences to clarify this*

- l.167-169: "CLM4.5 etc": this information does not seem relevant at this point. Maybe put this at the end of the paragraph and in the overall comparison that I suggest above.

*We will add this information in the new table*

- l.178, l.185: "are also used": so that there are two sets of methane emissions by CLM5/JULES in this study?

*Yes, the other one from CMIP6 coupled simulations and other from stand-alone simulations*

- l.188: "recently": does this mean it is what is used in this study?

*Yes*

- l.195: "of which": does this mean the ensemble is larger but here a subset is used? If so, why keep in the sub-ensemble models which are targeted in the study? Explain why it is not an issue: maybe because the set-ups are different enough?

*We used all available data. Please see the first reply (marked with a star) under Materials and methods*

- l.196: "GCP-diag": the full ensemble or the subset?

*Full ensemble*

- l.197: "8 models": same remark about the models which are part of the ensemble and targeted in this study.

*Please see the first reply (marked with a star) under Materials and methods*

- l.200: "inversion models": this does not look like the right name for these: any (chemistry-)transport model can be embedded in an inversion framework.

*We will change to: inversion frameworks*

- l.203: "the share": what does it mean?

*We will change to: 'the wetland proportion of the total flux*

- l.203-208: "In addition to GPC...": what is the logical link/relevancy of these sentences to the previous explanation? Re-order the description of the inversions to include the information on the priors where the reader can understand which priors go into which set of inversions.

*We will re-order the sentences*

- l.217: "processed and analysed on a 1x1 degree grid": how? Is this expected to lead to some issues?

*See below*

- l.218: "remapped by bilinear interpolation onto 1x1 degree grid": is this different from the regridding of previous sentence?

*Bilinear remapping has only been done when it was necessary to change resolution. Sentences will be rearranged for clarity: The JSBACH-H, LPX-Bern and GCP-prior results were remapped by bilinear interpolation onto 1x1 degree grid for use in CTE-CH4 atmospheric inversions. Analyses of flux results including ecosystem process model results, atmospheric inversion results and up-scaled eddy fluxes was made on a 1x1 degree grid.*

- l.224: "mostly": but not all?

*In addition to Obspack v2.0, there are few stations in the study region that are not part of Obspack. They are presented in Figure 1 and will be listed in a new table.*

- l.231: "were better constrained over that larger region than 1x1 degree given the limited number of surface stations": probably not very clear for readers who are not into flux inversion... Maybe simply state that the uncertainty on the retrieved fluxes is too large at the pixel's resolution but is satisfying when aggregating over the whole region.

*We will modify the text accordingly*

- l.231-235: "In Northern...": put the description of the inputs of the inversion before the description of the results i.e. the posterior (or retrieved) emissions.

*We will move the sentence*

- l.242-243: "grid-wise mean of the three emission maps available for years 2013 and 2014.": why this choice?

*The grid-wise mean of the maps was created to have one robust map given the large variability of individual wetland maps. On reflection, it would have been possible to use all three as separate maps and perhaps have some range of variability in the results for the seasonality of fluxes.*

- l.252: "significantly": define.

*We will reformulate the sentence*

- l.253-254: "creating confidence in the validity of the CRU-JRA climate data approach": not clear: to me, it shows that the uncoupled approach is OK for the targeted scientific question but "confidence in the validity of the approach" is too strong, it may give the idea that the coupling is not necessary at all. Please also define confidence in this context.

*We will reformulate the sentence*

- l.254-256: "In general, CRU gridded datasets are found to be suitable for vegetation analyses and well comparable to e.g. MERRA-2 and ERA5-Land reanalysis datasets, performing well even in remote areas with few observations (Zandler et al., 2020)": put in the description of CRU? The order in which the items of information are presented in the text is very confusing for the reader. Please review the whole structure of the sections to clearly first present the inputs and then their use and finally comment on the impact on methane emissions.

*We will move the sentence close to CRU description*

- l.258: "as their mean temperatures were always above zero": why not have a look at negative temperatures? It looks like April is also positive, according to Fig. 2.

*When the ground is frozen, there are very few methane emissions in the process models and in the biospheric inversion component, and therefore the spring months with frequent negative temperatures would not add much new information to the analysis.*

- l.261: "those growing season months": = May to Oct? Please define the growing season in reality vs in the models.

*The growing season can be defined in many ways. A traditional way is to define the growing season using air temperature but ecosystem data, such as measured CO2 fluxes or bud burst (greening) data can also be used to define the growing season. The length of the growing season also varies considerably when moving from south to north in the study region. In the southernmost parts of Finland it is on average 185 days long while in the northernmost parts it is 105 days long*

*according to the Finnish Meteorological Institute's temperature sum -based definition. We will add a clarification of the temperature sum -based growing season in the text.*

- l.263: "correlating": do you mean to use here "correlate"?

*We will change to: correlate*

**Results**

Same general remark on the order in which the information is delivered: it must be revised.

Section 3.2: I don't understand this section at all. The first paragraph announced work on two models but it's not what is done in the following paragraphs. The messages are not clear at all, the text is too descriptive and lacks synthesis/clear messages. I think the whole section is to be built again/rewritten.

*We will reorganize the section, start with all model results and then continue with the two models with contrasting temperature and precipitation dependencies*

Figure 3: not very easy to read with a different color scale for each model. At least try to have only two of them e.g. one for the large emitters and one for the smallest ones. And have them be easy multiples of one another. Another solution: plot some indicator, normalized, without unit, that would be in the same range for all models. It is always possible to put the details with tailored scales in the supplementary material. Show on the panels, e.g. with lines or rectangles, the regions of high temperatures, low precipitations, etc, which are used in the text.

*We will revise the figure with a normalized indicator and add markers or lines for specific regions.*

Figure 4: same remarks as for Fig. 3

Figure 6: it is not clear what GCP-post is, the ensemble?

*It is the posterior optimized estimate from CTE-CH4 inversion model using GCP-prior. We will clarify this in the figure*

Tab.S1: multiplier = ratio?

*We will change to: ratio*

Figure S3: should probably be used in the rewriting of section 3.2 as it seems very informative about the maximum of the seasonal cycle...

*We will refer to the results in the figure*

- l.271: "Natural wetland fluxes, including those from peat-lands and wet and dry mineral lands as well as inundated lands": list of types of wetlands is required at the beginning of the paper!

*We will list the broad types considered in this paper, see earlier comments*

- l.272: "below for": what does it mean?

*That the **growing season temperature and precipitation responses of wetland fluxes** are studied below*

- l.274-275: "the temperature and precipitation responses": how can this change in the posterior if it's flux inversions? The correlation change because the post emissions change but nothing is said by flux inversions on the relations between temperature/precipitations and emissions (see also previous remarks).

*We will reformulate the sentence*

- l.275-276: "The seasonal cycle is also compared to up-scaled eddy covariance flux observations.": please re-order, this information should not be at the end of this paragraph.

*We will re-order*

- l.279: "highest emissions": show on the color scale what are the highest emissions: the 10th highest percentile%?

*We will modify the figure by adding lines as explained earlier*

- l.279: "high temperature": show on the axis what high temperatures are.

*We will modify the figure by adding lines as explained earlier*

- l.278-282: what is the scientific message of this paragraph?

*The message is that the models have varied responses to temperature and precipitation (mentioned in the beginning of the paragraph)*

- l.285: "significant": define.

*We will add the proportion of the variance explained (now in figure) to the text.*

- l.285: "generally weaker": precise how much, how often?

*We will add the proportion of the variance explained (now in figure) to the text.*

- l.287: "Multiple regression...": put in the description of the method, not in the results.

*We will move the sentence to description of the method*

- l.284-288: what is the message on these results? What does it mean we must study in the following?

*We found that models such as JSBACH and LPX-Bern had very dissimilar temperature and precipitation responses. Because of this, we used those two models as priors in the inversions in the following. We will add this to the text.*

- l.290-291: Already stated before, remove from here.

*We will modify the sentence to make a better link to the previous paragraph.*

- l.292: "In total": = over the whole region during the whole period?

*Yes*

- l.293: "bringing the flux estimates closer together": maybe a bar plot would be useful to display these results.

*There is already Suppl Fig S2 showing the change from prior to posterior; we consider that adding another plot would not bring essential information*

- l.298: "multipliers": = ratios?

*We will change to: ratios*

- l.299: "above 92%": why 92? Why not 90?

*See below*

- l.300: why 64%? Is there an idea of Gaussianity somewhere?

*These numbers are all related to posterior/prior ratio of 2.0, which was defined as high increase from prior.*

- l.300-302: "The highest increase was proposed for July 2014 with second highest mean temperature of 16.7 °C. However, the July 2018 record high heatwave with mean temperature of 17.2 °C was not among the highest posterior increases.": This suggests that July 2014 is not well represented by the models (prior way too low) whereas July 2018 is (prior already OK so no large correction after inversion). The question then arises of what causes this difference in the models? What do they capture right in 2014 and wrong in 2018? Is it linked to the temperatures or to another parameter that differs between the two heat waves?

*In LPX-Bern the fluxes are small and on average the fluxes are increased from prior to posterior. Thus it could be said that the model is generally not doing well, because the fluxes need to be increased, although the same could be said about all models, since the inversion almost always changes the fluxes. However, the LPX-Bern fluxes in 2018 are not increased as much in posterior as in the other years, and this may not be because it is doing well in that year, but rather because it is not producing the drought well enough and therefore it stays at a higher level and the increase is*

*smaller in the posterior fluxes. However, it is not clear how much the drought actually impacted methane fluxes at the regional scale and whether the model should have produced drought everywhere, as e.g. Rinne et al (2020) found that not all wetlands suffered from drought. We will expand on this discussion in the text.*

- l.302: "43 mm in July": 2018? Is this part of the answer to the previous comment?

*yes*

- l.303-305: "Some of the highest...": I understand that some high precipitation emissions are too low in the priors: so what is wrong with them? Moreover, what about the rest of the high precipitation months?

*see above*

- l.308: "(above unity, i.e. above 88% percentile of all values)": what does this mean?

*The postrior/prior ratio is larger than one, which corresponds to the 88% percentile*

- l.308: "(51% percentile)": therefore in almost half the cases?

*yes*

- l.309: "significantly": how much?

*We will add the posterior/prior ratio*

- l.312: "relatively": how much is it?

*We will add the posterior/prior ratio*

- l.313: "2018 (and 2014 and 2006)": this lumps together different cases: 2018 ends up with average emissions but 2014 and 2006 end up with high emissions. And what about 2010 and 2005 regarding precipitations?

*We will separate the cases and expand the text.*

- l.315: "otherwise": I don't understand the logical link.

*It refers to other increases. We will split the sentence into two for clarity.*

- l.316: "July 2018 did not show high posterior fluxes here, while": consistent with JSBACH-H in the previous paragraph but the way it's stated suggests that it is a particularity of these simulations... Please clarify.

*We will clarify and add the reference to JSBACH-H*

- l.318: "balanced": what does it mean? What is a balanced / an unbalanced prior?

*Both temperature and precipitation explain the variance of fluxes.*

- l.318-321: "The temperature and...": the message is not clear at all. Cut up the long sentence into several shorter ones and re-formulate.

*We will reformulate the sentence*

- l.325: "and also the different model components.": what is meant?

*The peatland fluxes, inundation fluxes and mineral soil fluxes, explained in the next sentence.*

- l.325: "Generally": please clarify.

*It is common to several models. We will remove the word as it is not really needed.*

- l.325: "total": = the whole region during the whole period? Or do you mean wetland fluxes = fluxes from these various categories?

*Fluxes from these various categories*

- l.327: "LPJ-GUESS": should it be LPX-Bern here?

*Yes, we will correct this*

- l.330-334: How is this a result of this study? It looks more like an explanation of how LPX-Bern works.

*See below*

- l.336-339: "Same remark as the previous one. Maybe the separate explanations of each model should go in the suppl?"

*The months of maximum emissions are results of a regional simulation and are of scientific value. Such results can not be deduced by simply looking at the process descriptions and dependencies, but can only be seen after a regional simulation, after which it is also necessary to also explain the result by considering the role of different model components and explain how the components work together to produce such a result. Results for seasonal cycles are commonly used in model benchmarking. Without seasonal cycles this study would lose much of its value.*

- l.341-347: Very descriptive paragraph: what is the message?

*That the different methods suggested the flux maximum in August*

- l.349-353: what is the link of this paragraph with the rest of the section?

*This is a comparison of the other models and up-scaled eddy covariance fluxes. We will add a sentence about how these results compare to LPX-Bern and JSBACH-H.*

**Discussion**

I think there is an issue in the text with separating clearly what we know of the real world and what happens in the models' world(s). This makes the discussion unclear, particularly since what is known before hand and what is learned through the study is not clear (linked but not totally due to the issue about the times of the verbs).

- l.355: "According to process models": and not in the real world?

*This is learned in the study, and is therefore a result of the process modelling, not real observation.*

- l.356: "they comprise": what does it mean?

*they comprise = they include*

- l.358: "could be linked": we should know from how the models were built, shouldn't we?

*We will change to: is*

- l.359-360: "Precipitation has a dual role: it presumably increases the wetland area by wetting dry upland soils and raises the water table in the permanent wetlands": in the real world or in the models or both?

*We will change to: According to models and observations, precipitation has a dual role:*

- l.360: "constant/neglecting": what is meant? That it's as bad to use a constant value or to neglect the process?

*We will change to:*

In models, weak precipitation constraint could arise from using a constant value or completely neglecting the wet mineral soil emissions or from maintaining static proportions of wet and dry mineral land area over the growing season

- l.363: "being at largest after prolonged precipitation and generally in autumn (September-October) when the evapotranspiration had already decreased from high growing season levels.": in the model's or in the real world?

We will add: modelled evapotranspiration

- l.364-370: "According to observations...": what is the link with the discussion on the models' results in this study?

*We will add:*

*This suggests that the model should have drier soils in the autumn and the modelled wet mineral soil emissions should have an earlier maximum.*

- l.372-381: I guess the idea of this paragraph is to compare the findings of this study to others but it is not made clear at all because of the order in which the information is given (and the times of the verbs). Please reorder and reformulate.

*This paragraph only reviews the findings of others, not this study*

- l.372-374: "A modeling study by Poulter et al. (2017) concluded that in boreal regions CH 4 emissions were best correlated with wetland area, followed by temperature and precipitation (as applied with one-month delay). However, methane emissions were highly correlated with temperature in some models (e.g. JULES) which had a high temperature sensitivity.": does this mean this study is not in agreement with Poulter's? What would this imply?

*This paragraph only discusses results found in the literature. Our study is in agreement with Poulter et al., who studied the same models but with different set-ups, including JULES. We also found that JULES is sensitive to temperature*

*We will add:* *However, according to Poulter et al. (2017),* *methane emissions are highly correlated with temperature in some models (e.g. JULES,* *similar to our study**)*

- l.374: "In general": please be more precise.

*Also from Poulter et al*

- l.376-377: "noting that the co-limitation of temperature and precipitation would emerge for the more southern climate zones.": what is the link with this study?

**In our study we found co-limitation also in boreal zone. We will add this in the text**

- l.383: "towards August": instead of July?

*Instead of July or September, depending on the model*

- l.387-388: "In our work, many models had their seasonal maxima in July or August, notable exceptions being CLM4.5 and CLM5 (bias towards spring) and LPX-Bern (bias towards autumn).": put in the first sentence, where the results of the study are summarized (prior to comparison to the literature).

*We will consider changing the order of the sentences. As it is now, the paragraph starts with a discussion of our inversion results, and continues with our process model results. The reference to Warwick et al is placed in between and binds these results together as they report an improvement in mixing ratios because of a change in ecosystem model fluxes. Those results support our findings. Changing the order would break the link.*

- l.388-390: "The dominance of mineral land over peatland emissions may delay the month of the maximum emissions, as well as using a large wetland extent in late summer. ": is this the reason why the models give a maximum in July instead of August?

*If the model is dominated by peatland emissions then it indeed gives an earlier maximum*

- l.390-391: "Placing more peatlands in the southern parts of the region (like in GCP-diag) or having a weak temperature response could bring an earlier and longer seasonal emission maximum.": is this a suggestion of how to modify models?

*No, it is a consideration of possible reasons for an early emission maximum.*

*We will change to: may possibly result in*

- l.391-392: "A pronounced inundation period after snow melt could induce large methane emissions in spring.": in the real world? What is the link to the previous and next sentences?

*This refers to CLM early spring maximum and possible reasons for that. We will add: like in CLM4.5 and CLM 5*

- l.393: "from satellite observations": not clear to me. Is this an explanation of why the inversions put the maximum in August? But they don't assimilate satellite data. I don't understand the link between this piece of information and the rest of the paragraph.

*It is linked to the sentence just before the one about large methane emissions in spring. Large spring emissions could be caused by an early start of photosynthetic activity, which provides substrate for methane production. In particular, if we have an early spring with warm temperatures, wet soils from snow melt, deep unfrozen soil layers, and an early start of vegetation photosynthetical activity, then we may also have and early start also in methane emissions. Modelled vegetation phenology may be biased so that the model (in average) has too early start of photosynthetic activity. However, satellite observations do not support this.*

*We will add:*

growing season was usually delayed from satellite observations in northern latitudes, which means that there are other reasons for the early methane emission maximum than the early start of model photosynthesis.

- l.393-394: "According to flux measurements at boreal peatlands, the month of highest emissions was July or August depending on the year (e.g. Rinne et al., 2020).": does this mean that models put

the maximum in July, inversions in August but actually, we don't know which is closer to the real world?

*There is variation from year to year, but if we take an average over many years we can tell what is the maximum. The challenge is that we do not have too enough sites in the region and/or they do not cover enough years to be able to determine the maximum month with high confidence.*

- l.399: "anomalous": please define.

*not always anomalous = did not have high fluxes*

- l.402-403: "Rinne et al. (2020) also noted that methane emissions in four out of five Fennoscandian wetland sites were decreased in 2018 due to a decrease in water table levels. The summer months with high precipitation often resulted in high posterior emissions.": is this put here to suggest that the inversions are consistent with independent measurements? Shouldn't this be stated when presenting the inversions as a reference for comparison?

*Inversions are presented in the previous sentence: 'In the (inversion) posterior and especially in July 2018 the fluxes were decreased, possibly because of the decrease in soil water table level as June and July 2018 suffered from a lack of precipitation (see e.g. Peters et al., 2020)'*

*We will add: methane emissions in four out of five Fennoscandian wetland sites were decreased in 2018 due to a decrease in water table levels, suggesting that many wetlands suffered from drought, which may be reflected in regional results.*

- l.404-405: "The year 2011 with observed high methane emissions from upland soil in northern Fennoscandia (Lohila et al., 2016) did not stand out in posterior emissions": this suggests that the models do not well represent the high precipitation months. Why?

*It is possible that the signal observed by Lohila et al was not seen in the larger region of this study, but only in northern Fennoscandia.*

*We will add this in the text*

- l.404-406: "The year 2011 with observed high methane emissions from upland soil in northern Fennoscandia (Lohila et al., 2016) did not stand out in posterior emissions, but large increases were assigned to high precipitation periods in e.g. August 2008, 2016 and late summer 2007": is this the same message as the sentence "The summer months with high precipitations..."?

*We will change the order of the sentences to make the message clearer.*

- l.406-407: "August was also the month of the average seasonal precipitation maximum, while the average temperature maximum was in July.": this sentence is the last in the paragraph. This suggests to the reader that it is very important. Nevertheless, it is actually only a description of the input data of temperature and precipitation, to state when their max is. Please, reorder the logical "progression" of the sections/paragraphs to provide the information in a logical progression.

*We will change the order of sentences in the last paragraph, and add a small summary.*

**Conclusion**

The conclusion depends on how well the introduction presents the challenges tackled by the study. Therefore, in the current state of the text, seemingly trivial messages appear, e.g. that "it is important to study the overall responses of the emissions to air temperature and precipitations". The reader may winder that it is not already done and if this might due to the fact that models take only C stocks and water table depth into account. The conclusion will have to be re-written after taking into account the comments on all the previous sections.

- l.412-413: "agreed on the month of maximum emissions": this is not clear from the discussion.

*We will clarify this in the discussion*

- l.413: "balanced": must be defined, see previous remarks.

*We will add: both temperature and precipitation explain the variance of fluxes,*

- l.413: "significantly": see previous remarks.

*We will change to: was important*

- l.415: "to move emissions of both in posterior towards co-limitation of temperature and precipitation.": not clear, please reformulate.

*We will clarify the sentence*

- l.415-417: "The set-up of different emission components (peatland emissions, mineral land fluxes) had a significant role in building up the response patterns": how is this not trivial? What is the link with the inversions

*We will add the link with inversions; a short summary of results*

- l.419: "multi-year average response patterns": please explain.

*Where data from several years was used to determine the responses. We will add this to the text.*

- l.420: "anomalous cases of severe droughts with significant water table drawdown in pristine peatlands": not obvious from the discussion. Pristine wetlands, "anomalous" and "significant" not defined in the text.

*We will change to: the cases of severe droughts leading to water table drawdown in wetlands*

- l.420-421: "corresponding reductions in methane emissions": Corresponding reductions are simulated by the models? The sentence seems to lack a verb.

*We will change to: leading to reductions…*

- l.421-423: "Depending on the model, wet mineral soil and inundated land emissions can modify the seasonality of methane emissions together with peatland emissions. Therefore, it is essential to pay more attention to the role of the individual emission components, their magnitude, annual cycle and spatial extent in different regions": how is this not trivial?

*This is not trivial as peatland emissions are considered to be the main component of methane emissions and wet mineral soils are often not considered to be as important. However, despite the lower emission per unit area, the regions emitting methane can be very large and therefore alter the seasonality of the regional emissions. Furthermore, the wet mineral lands are implemented in different ways in the models and can therefore produce different outcomes for the total flux.*

*We will add text about this in the conclusions.*

- l.423-424: "how the fluxes should be scaled up from site to region": isn't it how people working on process model work? What does this study suggest they change in their work?

*When scaling up from site to region the area of the mineral soils, peatlands, lakes, rivers etc must be taken into account, as discussed in the introduction. In the boreal zone it is challenging to accurately define the extent of the different landcover classes that contribute to methane emissions, and when adding up different layers, there is a risk of erroneous or double counting of the emissions. Therefore, process models should use harmonized land cover datasets, and if the extents of wet mineral lands and peatlands are simulated by the model, validate the estimate against satellite and /or ground based inventory datasets. The models should also use process-based descriptions for the mineral and peat soil fluxes to simulate the flux responses to climate drivers.*

*We will add text about this in the conclusions.*

**Technical corrections**

- check the acronyms: they need to be fully explicit only at their first occurrence.

*We will check the acronyms*

- check the language: consistency between US/GB (s/z), accurate vocabulary outside of the strictly scientific (utilise/use) and the time of the verbs.

*We will check the language*

---

## Author Response (AR1)

*We thank the reviewers for the detailed and constructive comments. We agree with most of the suggestions and will make corresponding changes. We hope that our answers are sufficient and believe the manuscript will be much improved after the corrections*

**Reviewer 1**

The authors present an assessment of the impact of temperature and precipitations on methane emissions by Northern European wetlands as simulated by 6 process models. This study makes use of a set of ecosystem models, atmospheric inversions and fluxes deduced from eddy-covariance measurements to gain insights on the behavior of the 6 models with regard to temperature and precipitations.

**General comments**

The protocol of the study, with the various inter-comparisons, is very good, making use of all possible/relevant ways to assess the behavior of the 6 targeted models. It will be very useful to a wide community, working on process models but also on other models and even atmospheric inversions.

Nevertheless, because it should be usable by people with different backgrounds and probably also due to the many co-authors from different approaches, the structure of the manuscript, the order in which the ideas and information is delivered must be revised. Inside many sections and subsections, there are missing links between the ideas or pieces of information. This is not helped by the use of verbs in the past tense which should probably be revised by a native speaker. Even though I am not a native speaker, I was confused by this point about which information is from what was done before the study or not in the scope of the paper, what has been done for the study during the preparatory phase and what has been learned from the study.

*Here we kindly note that the co-authors include three native speakers who revised the manuscript and corrected the language.*

**Specific comments**

**Abstract**

I think the abstract may be rewritten to better represent the quality and important results of the study: some key sentences are too vague or general, as detailed below.

- the whole abstract: why not write in the present tense?

*We changed to present tense*

- l.31: "compared to multi-model mean": a multimodel mean = one reference?

*There is more than one multi-model mean, we changed to multi-model means*

- l.34-35: "how the inversion attempts to change the prior fluxes in the posterior": clumsily stated, not very clear for readers outside inversions. Please rephrase, possibly on the lines of how the assimilation of atmospheric data correct the fluxes.

*We changed to:*

how the *assimilation of atmospheric concentration data changes the flux estimates*

- l.36: "to move emissions": i.e. some of the emissions or all emissions?

*We changed to:*

*wetland* emissions

- l.36: "in posterior": this is probably too specific a word for an abstract of a paper not aimed at the inversion community. Please rephrase.

*We removed the word*

- l.36-37: "in general", "often": this is too vague, please quantify or at least, indicate in space, in time...

*See below*

- l.37-38: "This was not the case for the warm and dry period of summer 2018": with in general and often in the previous sentence, this sentence does not bring any information.

*We changed to:*

*Between 2000 and 2018, periods of high temperature and/or high precipitation often resulted in increased emissions. However, the dry summer of 2018 did not result in increased emissions despite the high temperatures.*

- l.39-40: "varied from May to September": from one year to another? From one model to another? Please clarify.

*We changed to:*

*The month with the highest emissions varies from May to September among the models.*

- l.41:"balanced": what is a balanced response? Please define or rephrase.

*We changed to:*

**However, multi-model means, inversions and up-scaled eddy covariance flux observations** *agree on the month of maximum emissions and are co-limited by temperature and precipitation*

- l.42,43: "significant":  define/quantify what is significant in the context. Or avoid this word outside a clear statistics framework.

*We changed to: important*

- l.43-44: "it is essential to pay more attention to the magnitude, composition, annual cycle and climate driver responses of wetland emissions in different regions." I guess everybody working on process-models knows about this. Isn't there a more practical or precise conclusion from this work?

*We revised the wording of the sentence to be more precise: Considering the significant differences among the models, it is essential to pay more attention to the regional representation of wet and dry mineral soils and periodic flooding which contribute to the seasonality and magnitude of methane fluxes. The realistic representation of temperature dependence of the peat soil fluxes is also important. Furthermore, it is important to use process-based descriptions for both mineral and peat soil fluxes to simulate the flux responses to climate drivers.*

**Introduction**

As such, the introduction does not make it very clear how models use input information and what the issues are with temperature and precipitations.

'Clarification on how models use input information':

*L46 We changed to:*

*Temperature, soil moisture, water table depth and primary production drive the carbon accumulation, respiration and methane emissions from peatlands* *and are modelled by ecosystem process models using atmospheric climate data such as temperature and precipitation as input to the simulations*

'Clarification on issue with temperature and precipitation':

*We assume the issue refers to clarifying the temperature and precipitation responses in a regional context, as this article is not addressing the calibration of individual models at specific sites and therefore is not dealing with any specific issue with calibration of temperature or precipitation responses, just diagnosing the characteristics of the models when applied to estimation of regional CH4 fluxes. We address the importance of regional temperature and precipitation studies in the following:*

*L68-73…The model predictions of regional annual cycles of methane emission differ significantly and the future estimates of the total global methane emissions are highly variable (Stocker et al., 2013, Saunois et al., 2020). Therefore it is useful to take a more climate-oriented perspective to the drivers of the methane emission in order to make better predictions of the responses to future climate change (Koffi et al., 2020), and to emphasize the regional approaches. It is important to study the responses of the emissions to air temperature and precipitation, as it defines the response of wetland emissions to climate change.*

Regarding the order in which the pieces of information are delivered, first, a presentation of the various types of wetlands in the real world would be useful - it is done partly in the Results, when listing what types are taken into account in some models; then, a presentation of what is done in the models (for example, ignore lakes).

*Due to complexity of wetlands, there are various definitions for different types of wetlands, and they vary from one region or country to another. Harmonization of the definitions and creation of a consistent wetland map is a topic of more than one on-going projects. Furthermore, the global ecosystem models are not presenting the different types in detail. Therefore we think presenting wetland types in broad categories, which are simulated by the models, are sufficient for this study. We present in the introduction (first paragraph) the different types of wetlands in our models; water-logged peatlands, wet mineral lands and inundated lands.*

*To further clarify, we added (l46):*

*Wetlands considered in this study include those at peatlands and mineral lands as well as periodically inundated, i.e. flooded lands.*

Overall, is not made clear enough what happens in the real world and what happens in the models, so that it's difficult to understand the challenges this study deals with. Examples: it is difficult to understand why the emissions of process models have not already been validated regarding temperature and precipitations dependencies;

*The process models are often validated against local measurement station data and different models are validated against different local data sets. Here we take a regional approach which is sought for in recent literature and is necessary for global studies, and run several models for the same region, upscaling local fluxes to a regional flux. The temperature and precipitation dependencies may differ in the up-scaled regional fluxes because of differences in model set-ups, calibration and validation, and presence of the different wetland types and distributions in the region. We address the challenges related to regional fluxes in the introduction and references therein (see also above ' issues with temperature and precipitation').*

the way inversions can inform process models is presented in a misleading way: too optimistic if it's flux inversion, not clear if it's process models' parameters which are inverted.

*Here we are looking at the regional up-scaled fluxes and responses. We do not claim that inversions can inform process models on their responses at site level or optimize their parameters. It is also true that the results from atmospheric inversion is not a ground truth. However, for regional fluxes atmospheric inversions provide more consistent results with atmospheric state (concentrations), and therefore, could be taken as one of very few means for evaluation of process*

*models. Although atmospheric inversion does not provide direct information about how process models' parameters or climate responses should be improved, we argue that the evaluation with inverse models could give a guidance towards which the process models should be improved.*

*We modified the text as follows:*

*Atmospheric inverse models rely on atmospheric methane concentrations, and they provide a top-down view of the methane emissions. Their results can be used to study responses of regional methane emissions to climate drivers. In process models the responses are more subject to how the processes were built and dependencies constructed, and how the fluxes were up-scaled. Therefore, it is worthwhile comparing the atmospheric inversion models to process models and study how their regional emission estimates and climate responses differ.*

- l.46: "the second most important greenhouse gas": ANTHROPOGENIC!

*We removed the sentence*

- l.48: "peatlands": please list the types of wetlands before going into details for each of them.

*We added:*

*Wetlands considered in this study include those at peatlands and mineral lands as well as periodically inundated, i.e. flooded lands.*

- l.52: "There are accurate peat-land maps": what about the mineral lands, mentioned just before but never again discussed?

*We changed the order of sentences in the paragraph to make the discussion of mineral lands more connected, and added sentences:*

*There are accurate peatland maps for the northern regions based on in situ data of peat layer thickness (e.g. Xu et al., 2018, Tanneberger et al., 2017), which enable estimations of the peatland methane emissions by process models if the soil water table level and soil carbon processes providing substrate for methane production are well represented. Land not covered by peatlands includes mineral land. In addition, mineral lands can act as a source of methane if the soil is very moist or inundated (Lohila et al., 2016, Wolf et al., 2011, see also Bansal et al., 2023), with a significant contribution from the organic layer on top of the soil. The soil moisture and land inundation can also be estimated by models together with peat accumulation, though it is still challenging (e.g. Loisel, et al., 2021, Ito et al., 2020). Soil moisture is an important input variable for mineral soil emission modelling (e.g. Curry et al., 2007).*

- l.57: "this feature is badly represented": this seems contradictory with the "accurate peat-land maps" of the previous paragraph. Explain more clearly what is know accurately and what is poorly known.

*Accurate peat-land maps are only one part of methane emission estimations; the soil hydrology (water table level) also needs to be simulated, and that is poorly represented. The extent of inundated lands is badly represented in the boreal zone because of forest canopy blocking the satellite view and lakes and ponds being interpreted as inundated land, as mentioned later in the text*

*We added:*

*There are accurate peatland maps for the northern regions based on in situ data of peat layer thickness (e.g. Xu et al., 2018, Tanneberger et al., 2017), which enable estimations of the peatland methane emissions by process models, if the soil water table level and soil carbon processes providing substrate for methane production are well represented.*

- l.67: "significantly": by how much?

*We removed the word*

- l.68: "more climate-oriented": i.e. instead of using maps of the extent? Please explain more clearly.

*We changed to:*

*Therefore, instead of studying response to wetland extent, it is useful to take a more climate-oriented perspective to the drivers of the methane emission*

- l.69: "to emphasize the regional approaches.": what does it mean?

*We modified the sentence:*

*Therefore, instead of studying response to wetland extent, it is useful to take a more climate-oriented perspective to the drivers of the methane emission in order to make better predictions of the responses to future climate change (Koffi et al., 2020). Further, it is important to emphasize the regional approaches as the drivers of emissions vary widely in their spatial distribution, climate and ecosystem type (Stavert et al., 2020)*

- l.70: "to study the responses of the emissions to air temperature and precipitation": the process models are built to take temperature and precipitation into account, aren't they? Haven't they be validated on these points? Make the challenge(s) clearer!

*As noted earlier, they are not validated by their regional responses. We clarified the text:*

*It is important to study the responses of the regional emissions to air temperature and precipitation, as it defines the response of regional wetland emissions to climate change.*

\- l.74: "arises": is this the right word?

*We changed to:*

*becomes apparent*

\- l.75: "inundation": please define.

*We changed to:*

*periodical inundation, i.e. flooding*

\- l.81: "according to atmospheric inversion modeling": this is too vague. Atmospheric inversions as such are not able to point at the causes/drivers of the corrections they apply to fluxes. So probably something more has been done than raw inversions by Thompson et al.

*We added:*

*and analysis of the results using climate reanalysis data (Dee et al., 2011).*

\- l.85-86: "provide a top-down view of the responses of methane emissions to climate drivers, attempting to detach them from the underlying prior assumptions.": not clear, even for somebody working on flux inversion. What is inverted here? Methane fluxes? If so, insights on climate drivers are not easy to obtain and the prior assumptions play a large part!

*We clarified the text:*

*Atmospheric inverse models rely on atmospheric methane concentrations, and they provide a top-down view of the methane emissions. Their results can be used to study responses of regional methane emissions to climate drivers.*

\- l.87-88: "atmospheric inversion models can be used to inform process models on how they should improve their emission estimates and climate responses": not so simple, if it's flux inversions: they can indicate where fluxes should be larger or smaller than what the process models compute but neither why they are too small or too large nor how to correct the process models. This last point is provided by inversion not of fluxes but of parameters of the process models. Please be very precise on the inversions you are dealing with.

*It is true that atmospheric inversion models do not directly inform why fluxes from process models are too small/large (e.g. whether it is due to drivers, bad parameter values or missing some processes), nor how to correct the parameters. Here, we meant to argue that the atmospheric inversion model results could also be studied with e.g. climate or ecosystem data and explain the relationships. Then the results about the relationships between fluxes and climate or ecosystem data can be compared to those from process models. Such comparison is worthwhile for informing process model to which direction they should improve their models. The inversion models themselves are not mechanistically explaining anything, but their flux results, based on assimilated atmospheric data, include the information needed to train a response model.*

*We changed to:*

*Therefore, it is worthwhile comparing the atmospheric inversion models to process models and study together how their regional emission estimates and climate responses differ.*

- l.89: "to study the responses of the emissions to air temperature and precipitation": I would think this is part of the evaluation/validation of process models! Please make the challenges clearer.

*We removed the sentence, as it somewhat repeated what was already said earlier.*

- l.91-92: "the ensemble of models from the Global Carbon Project (GCP) 2020 estimation": of which the Crescendo models are not part?

*See below (Materials and methods reply 1 marked with a star)*

- l.93-96: please cut up this sentence into several manageable shorter ones.

*We divided the sentence into shorter ones as follows:*

We use two of the models as well as the average of the GCP land ecosystem model ensemble (Saunois et al., 2020, Poulter et al., 2017) as priors of wetland emissions to inversions with Carbon Tracker Europe – CH4 (Tsuruta et al., 2017). We determine the sensitivity of the inversion to its prior and how this changes the interpretation of the flux responses to precipitation and temperature change in the boreal region in Fennoscandia.

**Materials and methods**

At least one model is in the ensemble used for comparison: please explain how this is not an issue for this study.

*\*The Global Carbon Project model ensemble provides a collection of best estimates from individual models. The models may share similar components so they are not fully independent. The varying*

*model set-ups may also produce different results from an earlier use case. Their predictions are not random but the probability to having systematic biases is smaller in comparison to a single model. The uncertainty of the model ensemble is considered to be smaller, or at least better known than the uncertainty of a single model, and as the size of a model ensemble increases the reliability of the result (may) also increase. To increase the size of an ensemble it is in our case necessary to include all models that have participated in GCP, also if they are variants of the same model like in the case of JSBACH, partly including same model components. The model estimates from Global Carbon project are used as a mean of several models, and they create a smoothed version of the temperature and precipitation responses. The individual model response patterns are reflected to the mean and finding the deviations is an important aspect of this work. Therefore we find it acceptable to use the full GCP model ensemble mean.*

At the end of each description, or better still, in an overall summary (maybe in a table), it would be very useful to have the limitations (e.g. neglected processes, poorly known inputs) and advantages of each model compared to the others (not need to list what they all do the same way) with regard to methane emissions. Maybe it would even be possible to state what is expected from each model e.g. performs better in a given region or season. I'd expect the impact of the spatial resolution to be large - but I don't run process models.

*We are comparing the models by their results in Discussion (line 360 ff): 'JSBACH-H, LPJ-GUESS and JULES were clearly more constrained by temperature. The reason behind this behaviour could be linked to strong temperature dependencies in the process descriptions (production, oxidation, transport)...'*

*We would like to be cautious in listing the limitations/advantages of the models in a deep process level and stating that some are better or worse than others, as we are studying regional temperature and precipitation responses which is only one aspect of model performance. For a full evaluation we would need to make a specifically designed benchmarking study. For example, we do not have the possibility to study the effect of spatial resolution, as we did not perform experiments with the same model set-up running in different resolutions. Additionally, this study partly uses published data, such as that from GCP, where the global limitations and advantages are discussed in detail. Furthermore, each model already has their own detailed description and calibration papers.*

*We extended the conclusions to give some general expectations for a model to perform well in the study region:*

*In order to perform well in the Fennoscandia region it is expected that a model will need to consider peatland and mineral land source and sink components of methane emissions and use up-to-date land cover description. If the peatland extent is simulated by the model, it needs to be validated against land cover data from bottom-up inventories.*

Please make the paragraphs of the various models homogeneous, with the same structure e.g. first the general history of development, then the relevant points for methane emissions and in the end, the sets of emissions used on this study with the name used in the remainder of the text.

*We rearranged the sentences in model descriptions from LPJ-GUESS and JULES part*

Subsections 2.1 to 2.7 : it is not clear how many runs or set-ups from each model are used in the study, please provide an overall summary, maybe as a table, with consistent tags. An assessment of the differences between the runs/set-ups and how they are expected to impact methane emissions and their response to temperature and precipitation is expected by the reader.

In several sections, the order of presentation is misleading i.e. the inputs are described after (part of) the description of the results. Please re-order.

*We rearranged the sentences in model descriptions.*

*We added a table (Table 1 below) describing the models and set-ups:*

*Table 1: Models and set-ups*

| Model | Type | Institution | Time period | Resolution | Reference |
|---|---|---|---|---|---|
| JSBACH-HIMMELI | Ecosystem process model | FMI, Univ. Helsinki, MPI | 2000 - 2018 | 1.875° x 1.875°, 0.1° x 0.1° | Kleinen et al. (2020), Raivonen et al. (2017) |
| LPX-Bern | Ecosystem process model | Univ. Bern | 2000 - 2018 | 0.5° x 0.5° | Lienert and Joos, (2018) |
| LPJ-GUESS | Ecosystem process model | Lund Univ. | 2000 - 2014 | 0.5° x 0.5° | Smith et al. (2014) |
| CLM4.5 | Ecosystem process model | NORCE | 2000 - 2014 | 1.25° x 0.9375° | Oleson et al. (2013) |
| CLM5 | Ecosystem process model | CMCC | 2000 - 2014 | 0.5° x 0.5° | Lawrence et al. (2019), Peano et al. (2021) |
| JULES | Ecosystem process model | UKMO, Univ. Exeter | 2000 - 2014 | 0.5° x 0.5° | Gedney et al. (2004), Comyn-Platt et al. (2018), Chadburn et al. (2020) |
| Carbon Tracker Europe – CH4 | Atmospheric inverse model | FMI | 2000 - 2018 | 1.0° x 1.0° | Tsuruta et al. (2017), Tenkanen et al. (2021) |
| Global Carbon Project (GCP) models: - GCP-diag - GCP-prog - GCP-prior - GCP-inversions | Means of - 12 diagnostic ecosystem models, - 8 prognostic ecosystem models, - climatological mean inversion prior - 5 atmospheric inverse models | GCP | 2000 - 2017 | Fluxes processed to 1.0° x 1.0° | Saunois et al. (2020), Poulter et al. (2017) |

| Up-scaled fluxes | Machine learning based random forest model | FMI | 2013 - 2014 | 1.0° x 1.0° | Peltola et al. (2019). |
|---|---|---|---|---|---|

Figure 1: provide information on stations, at least their full name and network.

*We added a table (Table S2 below) providing information on the stations:*

*Table S2. List of surface observation sites used in inversions  (see also Figure 1 )*

*Data type is categorized into two measurements (discrete (D) and continuous (C)). Date max is limited to 2018/12, which was the last month in inversions.*

| Sitecode | Site Name | Country | Contributor | Longitude | Latitude | Height | Data type | Date min [year/month] | Date max [year/month] |
|---|---|---|---|---|---|---|---|---|---|
| BAL | Baltic Sea | Poland | NOAA/ESRL | 17.22 | 55.35 | 28 | D | 1998/01 | 2011/06 |
| BIR | Birkenes | Norway | NILU | 8.25 | 58.39 | 219 | C | 2009/05 | 2018/12 |
| HTM | Hyltemossa | Sweden | ICOS-ATC, LUND-CEC | 13.42 | 56.10 | 265 | C | 2017/04 | 2018/12 |
| KJN | Kjolnes | Norway | Univ. Exeter | 29.23 | 70.85 | 20 | C | 2013/10 | 2018/12 |
| KMP | Kumpula | Finland | FMI | 24.96 | 60.20 | 53 | C | 2010/01 | 2018/12 |
| NOR | Norunda | Sweden | ICOS-ATC, LUND-CEC | 17.48 | 60.09 | 146 | C | 2017/04 | 2018/12 |
| PAL | Pallas-Sammaltunturi, GAW station | Finland | NOAA/ESRL | 24.12 | 67.97 | 570 | D | 2001/12 | 2018/12 |
| PAL | Pallas-Sammaltunturi, GAW station | Finland | FMI | 24.12 | 67.97 | 570 | C | 2004/02 | 2017/12 |
| PAL | Pallas-Sammaltunturi, GAW station | Finland | ICOS-ATC, FMI | 24.12 | 67.97 | 577 | C | 2017/09 | 2018/12 |
| PUI | Puijo | Finland | FMI | 27.66 | 62.91 | 84 | C | 2011/11 | 2018/12 |
| SMR | Hyytiala | Finland | ICOS-ATC, UHELS | 24.29 | 61.85 | 306 | C | 2016/12 | 2018/12 |
| SOD | Sodankylä | Finland | FMI | 26.64 | 67.36 | 227 | C | 2012/01 | 2018/12 |
| STM | Ocean Station "M" | Norway | NOAA/ESRL | 2.00 | 66.00 | 5 | D | 1999/01 | 2009/11 |
| SVB | Svartberget | Sweden | ICOS-ATC, SLU | 19.78 | 64.26 | 385 | C | 2017/06 | 2018/12 |
| UTO | Uto | Finland | ICOS-ATC, FMI | 21.37 | 59.78 | 65 | C | 2018/03 | 2018/12 |
| Stations outside study region | | | | | | | | | |
| CBW | Cabauw | Netherlands | University of Groningen | 4.93 | 51.97 | 199 | C | 2005/01 | 2013/06 |
| LIN | Lindenberg | Germany | ICOS-ATC, HPB | 14.12 | 52.17 | 171 | c | 2015/10 | 2018/12 |
| LUT | Lutjewad | Netherlands | ICOS-ATC, RUG | 6.35 | 53.40 | 61 | C | 2018/08 | 2018/12 |
| NGL | Neuglobsow | Germany | UBA-Germany | 13.03 | 53.17 | 68.4 | C | 1999/01 | 2013/12 |
| RGL | Ridge Hill | United Kingdom | UNIVBRIS - | 2.54 | 52.00 | 294 | C | 2012/02 | 2017/12 |
| TAC | Tacolneston | United Kingdom | NOAA/ESRL | 1.14 | 52.52 | 236 | D | 2014/06 | 2016/01 |
| TAC | Tacolneston | United Kingdom | UNIVBRIS | 1.14 | 52.52 | 241 | C | 2013/01 | 2017/12 |

| TER | Teriberka | Russian Federation | MGO | 35.10 | 69.20 | 42 | D | 1999/01 | 2018/12 |
|-----|-----------|--------------------|----|-------|-------|----|----|---------|---------|
| TOH | Torfhaus | Germany | ICOS-ATC, HPB | 10.54 | 51.81 | 948 | C | 2017/12 | 2018/12 |

- l.102: "utilised": is this the right word?

*We changed to used*

- l.104: "further developed in the recent H2020-CRESCENDO project": is there a reference for this e.g. a deliverable and/or report of the project?

*Project reports can be downloaded from https://cordis.europa.eu/project/id/641816/results*

- l.107: "in Global Carbon Project": the ensemble to use as a reference includes at least one of the 6 models targeted here: isn't is an issue?

*Please see the first reply (marked with a star) under Materials and methods*

- l.130: "was set to": based on what data/information?

*For improved peat accumulation rates (will be added in the text)*

- l.133: "manuscript": what does this mean? Currently a draft? Submitted?

*Tyystjärvi et al is in discussion in Biogeosciences (https://egusphere.copernicus.org/preprints/2024/egusphere-2023-3037/).*

*Li et al is in press in Tot. Sci. Env.( Li, X., Markkanen, T., Korkiakoski, M., Lohila, A., Leppänen, A., Aalto, T., Peltoniemi, M., Mäkipää, R., Kleinen, T., Raivonen, M., 2024. Modelling alternative harvest effects on soil CO2 and CH4 fluxes from peatland forests. Science of The Total Environment 175257. https://doi.org/10.1016/j.scitotenv.2024.175257)*

*We modified the status of the manuscripts in the text accordingly.*

- l.145: "suitability for peatland growth conditions": what does this mean?

*We will change to: a model that determines peatland growth conditions to simulate the peatland spatial distribution*

- l.158: "in general": do you mean it's not the case here? In some set-ups? Or not on some parts of the domain?

*We rearranged sentences to clarify this:*

*LPJ-GUESS land use is described by the Land Use Harmonization version 2 (Hurtt et al, 2020). WHyMe simulates methane production, three pathways of methane transport (diffusion, plant-mediated transport and ebullition) and methane oxidation. LPJ-GUESS-WHyMe stand-alone simulations for CRESCENDO project were made using a prescribed peatland map at a 0.5° resolution.*

- l.167-169: "CLM4.5 etc": this information does not seem relevant at this point. Maybe put this at the end of the paragraph and in the overall comparison that I suggest above.

*We added information about CLM 4.5 and CLM 5.0 in the new table. Otherwise, the order of sentences was not changed in order to keep the structure similar to that used for the other models.*

- l.178, l.185: "are also used": so that there are two sets of methane emissions by CLM5/JULES in this study?

*Yes, the other one from CMIP6 coupled simulations and other from stand-alone simulations*

- l.188: "recently": does this mean it is what is used in this study?

*Yes*

- l.195: "of which": does this mean the ensemble is larger but here a subset is used? If so, why keep in the sub-ensemble models which are targeted in the study? Explain why it is not an issue: maybe because the set-ups are different enough?

*We used all available data. Please see the first reply (marked with a star) under Materials and methods*

- l.196: "GCP-diag": the full ensemble or the subset?

*Full ensemble*

- l.197: "8 models": same remark about the models which are part of the ensemble and targeted in this study.

*Please see the first reply (marked with a star) under Materials and methods*

- l.200: "inversion models": this does not look like the right name for these: any (chemistry-)transport model can be embedded in an inversion framework.

*We changed to: inversion frameworks*

- l.203: "the share": what does it mean?

*We changed to:* *'the wetland proportion of the total flux'*

- l.203-208: "In addition to GPC...": what is the logical link/relevancy of these sentences to the previous explanation? Re-order the description of the inversions to include the information on the priors where the reader can understand which priors go into which set of inversions.

*We re-ordered the sentences:* *Wetland methane fluxes were extracted from the flux totals by the participating research groups and the wetland proportion of the total flux thus depends on the individual approaches chosen, and on the priors used. The wetland priors for the inversions were obtained from different sources. The WETCHIMP ensemble mean (Melton et al., 2013), or e.g. VISIT ecosystem model were used by the inversion models listed above'*

- l.217: "processed and analysed on a 1x1 degree grid": how? Is this expected to lead to some issues?

*See below*

- l.218: "remapped by bilinear interpolation onto 1x1 degree grid": is this different from the regridding of previous sentence?

*Bilinear remapping has only been done when it was necessary to change resolution. Sentences will be rearranged for clarity:* *The JSBACH-H, LPX-Bern and GCP-prior results were remapped by bilinear interpolation onto 1x1 degree grid for use in CTE-CH4 atmospheric inversions. Analyses of flux results including ecosystem process model results, atmospheric inversion results and up-scaled eddy fluxes was made on a 1x1 degree grid.*

- l.224: "mostly": but not all?

*In addition to Obspack v2.0, there are few stations in the study region that are not part of Obspack. They are presented in Figure 1 and are listed in a new table (Table S1).*

- l.231: "were better constrained over that larger region than 1x1 degree given the limited number of surface stations": probably not very clear for readers who are not into flux inversion... Maybe simply state that the uncertainty on the retrieved fluxes is too large at the pixel's resolution but is satisfying when aggregating over the whole region.

*We modified the text accordingly:* *'the posterior fluxes from the inversions were better constrained over that larger region than 1x1 degree, i.e. the flux uncertainty becomes very large in pixel resolution given the limited number of surface stations.'*

- l.231-235: "In Northern...": put the description of the inputs of the inversion before the description of the results i.e. the posterior (or retrieved) emissions.

*We moved the sentence.*

- l.242-243: "grid-wise mean of the three emission maps available for years 2013 and 2014.": why this choice?

*The grid-wise mean of the maps was created to have one robust map given the large variability of individual wetland maps. On reflection, it would have been possible to use all three as separate maps and perhaps have some range of variability in the results for the seasonality of fluxes.*

- l.252: "significantly": define.

*We reformulated the sentence (see below)*

- l.253-254: "creating confidence in the validity of the CRU-JRA climate data approach": not clear: to me, it shows that the uncoupled approach is OK for the targeted scientific question but "confidence in the validity of the approach" is too strong, it may give the idea that the coupling is not necessary at all. Please also define confidence in this context.

*We reformulated the sentence*

*The results did not change much in terms of placing the highest methane emissions in the temperature-precipitation space. Air temperature explained 76% of the flux variation in JULES and 71% in the coupled model run and 48% in CLM5 and 37% in the coupled model run. Precipitation explained less than 10% of the flux variation in all model runs (Suppl. Figure S1). It was therefore deemed appropriate to use the bias-corrected CRU-JRA data set in our analysis.*

- l.254-256: "In general, CRU gridded datasets are found to be suitable for vegetation analyses and well comparable to e.g. MERRA-2 and ERA5-Land reanalysis datasets, performing well even in remote areas with few observations (Zandler et al., 2020)": put in the description of CRU? The order in which the items of information are presented in the text is very confusing for the reader. Please review the whole structure of the sections to clearly first present the inputs and then their use and finally comment on the impact on methane emissions.

*We moved the sentence close to CRU description and clarified the structure of the section (see revised version in the text).*

- l.258: "as their mean temperatures were always above zero": why not have a look at negative temperatures? It looks like April is also positive, according to Fig. 2.

*When the ground is frozen, there are very few methane emissions in the process models and in the biospheric inversion component, and therefore the spring months with frequent negative temperatures would not add much new information to the analysis.*

- l.261: "those growing season months": = May to Oct? Please define the growing season in reality vs in the models.

*The growing season can be defined in many ways. A traditional way is to define the growing season using air temperature but ecosystem data, such as measured CO2 fluxes or bud burst (greening) data can also be used to define the growing season. The length of the growing season also varies considerably when moving from south to north in the study region. In the southernmost parts it is on average >220 days long while in the northernmost parts it is <100 days long (Aalto et al., 2022).*

*We added a clarification of the growing season in the text:*

*Length of the thermal growing season varies considerably when moving from south to north in the study region. In the southernmost parts it is on average >220 days long while in the northernmost parts it is <100 days long (Aalto et al., 2022).*

- l.263: "correlating": do you mean to use here "correlate"?

*We changed to: correlate*

**Results**

Same general remark on the order in which the information is delivered: it must be revised.

Section 3.2: I don't understand this section at all. The first paragraph announced work on two models but it's not what is done in the following paragraphs. The messages are not clear at all, the text is too descriptive and lacks synthesis/clear messages. I think the whole section is to be built again/rewritten.

*We reorganized the section (see text). We started with all model results and then continued with the two models with contrasting temperature and precipitation dependencies (see changes in text). We clarified the last sentences to make the message clearer:*

*… This indicates similar changes in the posterior regardless of the prior, i.e. the highest emissions were placed to August. The changes mostly took place in northern peatland areas with high methane emissions.*

Figure 3: not very easy to read with a different color scale for each model. At least try to have only two of them e.g. one for the large emitters and one for the smallest ones. And have them be easy multiples of one another. Another solution: plot some indicator, normalized, without unit, that would be in the same range for all models. It is always possible to put the details with tailored scales in the supplementary material. Show on the panels, e.g. with lines or rectangles, the regions of high temperatures, low precipitations, etc, which are used in the text.

*We added lines for high temperature and low precipitation regions in Figure 3. The model fluxes differ by their absolute magnitude. However, it is not the absolute flux levels per se that we are looking at here, but another characteristic: the temperature and precipitation dependencies. If we use common scale in the figures, the dynamic range of the colours becomes narrow in some figures because of the differences in flux magnitudes. The locations of the highest fluxes of the specific model in temperature - precipitation space become less clear. Therefore we chose not to change to a common scale. We hope that the added lines help in interpretation of the figure.*

Figure 4: same remarks as for Fig. 3

*We added lines for temperature and precipitation regions mentioned in the text in connection with Fig 4. We hope that the added lines help in interpretation of the figure.*

Figure 6: it is not clear what GCP-post is, the ensemble?

*It is the posterior optimized estimate from CTE-CH4 inversion model using GCP-prior. We modifed the texts in the legend to include CTE-CH4.*

Tab.S1: multiplier = ratio?

*We changed to: ratio*

Figure S3: should probably be used in the rewriting of section 3.2 as it seems very informative about the maximum of the seasonal cycle...

*We added further clarification in Section 3.2:*

*Comparing the change maps for northern Fennoscandia, the inversion with the JSBACH-H and LPX-Bern priors positioned the fluxes to a higher level in August, while in July the fluxes were placed at a lower level with respect to the seasonal mean adjustment (Suppl Fig S3). This indicates similar changes in the posterior regardless of the prior, i.e. the highest emissions were placed to August. The changes mostly took place in northern peatland areas with high methane emissions.*

- l.271: "Natural wetland fluxes, including those from peat-lands and wet and dry mineral lands as well as inundated lands": list of types of wetlands is required at the beginning of the paper!

*We clarified the listing of the broad types considered in this paper, see earlier comments for introduction*

- l.272: "below for": what does it mean?

*That the growing season temperature and precipitation responses of wetland fluxes are studied below*

- l.274-275: "the temperature and precipitation responses": how can this change in the posterior if it's flux inversions? The correlation change because the post emissions change but nothing is said by flux inversions on the relations between temperature/precipitations and emissions (see also previous remarks).

*We changed to: The results are analysed in order to examine how the inversions propose to change the prior CH4 emissions and how the correlations with temperature and precipitation, as well as the seasonal cycle of emissions change in the posterior*

- l.275-276: "The seasonal cycle is also compared to up-scaled eddy covariance flux observations.": please re-order, this information should not be at the end of this paragraph.

*We moved the sentence to Section 3.2*

- l.279: "highest emissions": show on the color scale what are the highest emissions: the 10th highest percentile%?

*We have added lines to the 75th highest percentile of flux values in Fig. 3.*

- l.279: "high temperature": show on the axis what high temperatures are.

*We have added lines to the 75th highest percentile of temperature and precipitation data in Fig. 3.*

- l.278-282: what is the scientific message of this paragraph?

*The message is that the models have varied responses to temperature and precipitation (mentioned in the beginning of the paragraph)*

- l.285: "significant": define.

*We added the proportion of the variance explained (previously only in Figure 3) to the text:*

**The regressions in Fig. 3 show the correlation of LPJ-GUESS, JSBACH-H and JULES emissions with temperature, indicating that the variance explained was significant** *as $R^2$ values for temperature were between 0.76 and 0.89 and P-values were < 0.01.*

- l.285: "generally weaker": precise how much, how often

*We added the proportion of the variance explained (previously only in Figure 3) to the text:*

**Correlations with precipitation were generally weaker, but still dominated over temperature in LPX-Bern** *(R2 for precipitation was 0.50),* **and CLM4.5** *(R2 for precipitation was 0.47).*

- l.287: "Multiple regression...": put in the description of the method, not in the results.

*We modified the sentence: According to least squares fit of flux results on a linear model with both temperature and precipitation as predictors, ai* *r temperature and precipitation could together explain at maximum 91% of the flux variation (JSBACH-H), but sometimes only 51% (CLM4.5).*

- l.284-288: what is the message on these results? What does it mean we must study in the following?

*We found that models such as JSBACH and LPX-Bern had very dissimilar temperature and precipitation responses. Because of this, we used those two models as priors in the inversions in the following. We added to the text:*

*Of the ecosystem models examined, temperature could explain most ($R^2_{Temp}$ = 0.84) of the JSBACH flux variance and precipitation a large part ($R^2_{Precip}$ = 0.50) of the LPX-Bern flux variance. Because of these contrasting features, these two models were chosen as priors in an inversion modelling experiment. The GCP-prior ($R^2_{Temp}$ = 0.45, $R^2_{Precip}$ = 0.35, see Fig. 4) was applied as a third prior for reference.*

- l.290-291: Already stated before, remove from here.

*We modified the sentence to make a better link to the previous paragraph:*

*LPX-Bern, JSBACH-H and GCP-prior were used as prior fluxes in the CTE-CH4 inversions for Fennoscandia.*

- l.292: "In total": = over the whole region during the whole period?

*Yes*

- l.293: "bringing the flux estimates closer together": maybe a bar plot would be useful to display these results.

*There is already Suppl Fig S2 showing the change from prior to posterior; we consider that adding another plot would not bring essential information*

- l.298: "multipliers": = ratios?

*We changed to: ratios*

- l.299: "above 92%": why 92? Why not 90?

*See below*

- l.300: why 64%? Is there an idea of Gaussianity somewhere?

*These numbers (92% and 64%) are related to posterior/prior ratio of 2.0, which was defined as high increase from prior.*

- l.300-302: "The highest increase was proposed for July 2014 with second highest mean temperature of 16.7 °C. However, the July 2018 record high heatwave with mean temperature of 17.2 °C was not among the highest posterior increases.": This suggests that July 2014 is not well represented by the models (prior way too low) whereas July 2018 is (prior already OK so no large correction after inversion). The question then arises of what causes this difference in the models? What do they capture right in 2014 and wrong in 2018? Is it linked to the temperatures or to another parameter that differs between the two heat waves?

*In LPX-Bern the fluxes are small and on average the fluxes are increased from prior to posterior. Thus it could be said that the model is generally not doing well, because the fluxes need to be increased, although the same could be said about all models, since the inversion almost always changes the fluxes. However, the LPX-Bern fluxes in 2018 are not increased as much in posterior as in the other years, and this may not be because it is doing well in that year, but rather because it is not producing the drought well enough and therefore it stays at a higher level and the increase is smaller in the posterior fluxes. However, it is not clear how much the drought actually impacted methane fluxes at the regional scale and whether the model should have produced drought everywhere, as e.g. Rinne et al (2020) found that not all wetlands suffered from drought. We expanded this discussion in the text:*

*The precipitation was record low, only 43 mm in July, which may explain the result, if the prior did not fully capture the possible drought effect and therefore the increase in the posterior was modest.*

- l.302: "43 mm in July": 2018? Is this part of the answer to the previous comment?

*yes*

- l.303-305: "Some of the highest...": I understand that some high precipitation emissions are too low in the priors: so what is wrong with them? Moreover, what about the rest of the high precipitation months?

*We added a sentence:*

*Some of the highest precipitation months like August 2008, 2016 and July-September 2007 with precipitation exceeding 100 mm, were already above average in the prior emissions but still experienced a large increase in the posterior. This could be because the inversion proposed an increase in the high temperature regime from the prior, and temperatures during these months were above 62% percentile of all values.*

- l.308: "(above unity, i.e. above 88% percentile of all values)": what does this mean?

*The posterior/prior ratio is larger than one, which corresponds to the 88% percentile*

- l.308: "(51% percentile)": therefore in almost half the cases?

*yes*

- l.309: "significantly": how much?

*Above 89% percentile (added to text)*

- l.312: "relatively": how much is it?

*We modified the sentence:*

*In the posterior the fluxes were decreased, but still the emissions stayed above 60% percentile of all values…*

- l.313: "2018 (and 2014 and 2006)": this lumps together different cases: 2018 ends up with average emissions but 2014 and 2006 end up with high emissions. And what about 2010 and 2005 regarding precipitations?

*We added:*

A decrease in soil water table may play a role, as July 2018 suffered from lack of precipitation. *The same was true for July 2014 and July 2006, but these months were not clearly distinguishable from precipitation-abundant months July 2005 and July 2010 based on posterior emissions.*

- l.315: "otherwise": I don't understand the logical link.

*We divided the sentence into two parts for clarity.*

- l.316: "July 2018 did not show high posterior fluxes here, while": consistent with JSBACH-H in the previous paragraph but the way it's stated suggests that it is a particularity of these simulations... Please clarify.

*We modified the sentence:*

*In addition, July 2018 did not show high posterior fluxes, aligning with the JSBACH-H results.*

- l.318: "balanced": what does it mean? What is a balanced / an unbalanced prior?

*Both temperature and precipitation explain the variance of fluxes.*

- l.318-321: "The temperature and...": the message is not clear at all. Cut up the long sentence into several shorter ones and re-formulate.

*We divided the sentence into two parts.*

- l.325: "and also the different model components.": what is meant?

*The peatland fluxes, inundation fluxes and mineral soil fluxes, explained in the next sentence.*

- l.325: "Generally": please clarify.

*It is common to several models. We removed the word as it is not really needed.*

- l.325: "total": = the whole region during the whole period? Or do you mean wetland fluxes = fluxes from these various categories?

*Fluxes from these various categories*

- l.327: "LPJ-GUESS": should it be LPX-Bern here?

*Yes, corrected*

- l.330-334: How is this a result of this study? It looks more like an explanation of how LPX-Bern works.

*See below*

- l.336-339: "Same remark as the previous one. Maybe the separate explanations of each model should go in the suppl?"

*The months of maximum emissions are results of a regional simulation and are of scientific value. Such results can not be deduced by simply looking at the process descriptions and dependencies, but can only be seen after a regional simulation, after which it is also necessary to also explain the result by considering the role of different model components and explain how the components work together to produce such a result. Results for seasonal cycles are commonly used in model benchmarking. Without seasonal cycles this study would lose much of its value.*

- l.341-347: Very descriptive paragraph: what is the message?

*That the different methods suggested the flux maximum in August*

- l.349-353: what is the link of this paragraph with the rest of the section?

*This is a comparison of the models and up-scaled eddy covariance fluxes. The paragraphs have been reordered to improve the flow of the text.*

**Discussion**

I think there is an issue in the text with separating clearly what we know of the real world and what happens in the models' world(s). This makes the discussion unclear, particularly since what is known before hand and what is learned through the study is not clear (linked but not totally due to the issue about the times of the verbs).

- l.355: "According to process models": and not in the real world?

*This is learned in the study, and is therefore a result of the process modelling, not real observation.*

- l.356: "they comprise": what does it mean?

*they comprise = they include*

- l.358: "could be linked": we should know from how the models were built, shouldn't we?

*We changed to: is*

- l.359-360: "Precipitation has a dual role: it presumably increases the wetland area by wetting dry upland soils and raises the water table in the permanent wetlands": in the real world or in the models or both?

*We changed to:* *According to models and observations, precipitation has a dual role:*

- l.360: "constant/neglecting": what is meant? That it's as bad to use a constant value or to neglect the process?

*We changed to:*

In models, weak precipitation constraint could arise from using a constant value or completely neglecting the wet mineral soil emissions or from maintaining static proportions of wet and dry mineral land area over the growing season

- l.363: "being at largest after prolonged precipitation and generally in autumn (September-October) when the evapotranspiration had already decreased from high growing season levels.": in the model's or in the real world?

We added: modelled evapotranspiration

- l.364-370: "According to observations...": what is the link with the discussion on the models' results in this study?

*We added:*

*This suggests that the model should have drier soils in the autumn and the modelled wet mineral soil emissions should have an earlier maximum.*

- l.372-381: I guess the idea of this paragraph is to compare the findings of this study to others but it is not made clear at all because of the order in which the information is given (and the times of the verbs). Please reorder and reformulate.

*This paragraph only reviews the findings of others, not this study*

- l.372-374: "A modeling study by Poulter et al. (2017) concluded that in boreal regions CH 4 emissions were best correlated with wetland area, followed by temperature and precipitation (as applied with one-month delay). However, methane emissions were highly correlated with temperature in some models (e.g. JULES) which had a high temperature sensitivity.": does this mean this study is not in agreement with Poulter's? What would this imply?

*This paragraph only discusses results found in the literature. Our study is in agreement with Poulter et al., who studied the same models but with different set-ups, including JULES. We also found that JULES is sensitive to temperature*

*We added:* *However, according to Poulter et al. (2017),* *methane emissions are highly correlated with temperature in some models (e.g. JULES,* *similar to our study)*

- l.374: "In general": please be more precise.

*Also from Poulter et al*

- l.376-377: "noting that the co-limitation of temperature and precipitation would emerge for the more southern climate zones.": what is the link with this study?

*In our study we found co-limitation also in boreal zone. We added:*

*Sensitivity of the boreal methane emissions to air temperature was confirmed by Koffi et al. (2020), noting that the co-limitation of temperature and precipitation would emerge for the more southern climate zones, whereas in our study co-limitation was found to be present also in boreal zone.*

- l.383: "towards August": instead of July?

*Instead of July or September, depending on the model*

- l.387-388: "In our work, many models had their seasonal maxima in July or August, notable exceptions being CLM4.5 and CLM5 (bias towards spring) and LPX-Bern (bias towards autumn).": put in the first sentence, where the results of the study are summarized (prior to comparison to the literature).

*As it is now, the paragraph starts with a discussion of our inversion results, and continues with our process model results. The reference to Warwick et al is placed in between and binds these results together as they report an improvement in mixing ratios because of a change in ecosystem model fluxes. Those results support our findings. Changing the order would break the link.*

- l.388-390: "The dominance of mineral land over peatland emissions may delay the month of the maximum emissions, as well as using a large wetland extent in late summer. ": is this the reason why the models give a maximum in July instead of August?

*If the model is dominated by peatland emissions then it indeed gives an earlier maximum*

- l.390-391: "Placing more peatlands in the southern parts of the region (like in GCP-diag) or having a weak temperature response could bring an earlier and longer seasonal emission maximum.": is this a suggestion of how to modify models?

*No, it is a consideration of possible reasons for an early emission maximum.*

*We changed to: may possibly result in*

- l.391-392: "A pronounced inundation period after snow melt could induce large methane emissions in spring.": in the real world? What is the link to the previous and next sentences?

*This refers to CLM early spring maximum and possible reasons for that.*

*We added: like in CLM4.5 and CLM 5*

- l.393: "from satellite observations": not clear to me. Is this an explanation of why the inversions put the maximum in August? But they don't assimilate satellite data. I don't understand the link between this piece of information and the rest of the paragraph.

*It is linked to the sentence just before the one about large methane emissions in spring. Large spring emissions could be caused by an early start of photosynthetic activity, which provides substrate for methane production. In particular, if we have an early spring with warm temperatures, wet soils from snow melt, deep unfrozen soil layers, and an early start of vegetation photosynthetical activity, then we may also have and early start also in methane emissions. Modelled vegetation phenology may be biased so that the model (in average) has too early start of photosynthetic activity. However, satellite observations do not support this.*

*We added:*

**growing season was usually delayed from satellite observations in northern latitudes,** *which means that there are other reasons for the early methane emission maximum than the early start of model photosynthesis.*

- l.393-394: "According to flux measurements at boreal peatlands, the month of highest emissions was July or August depending on the year (e.g. Rinne et al., 2020).": does this mean that models put the maximum in July, inversions in August but actually, we don't know which is closer to the real world?

*There is variation from year to year, but if we take an average over many years we can tell what is the maximum. The challenge is that we do not have too enough sites in the region and/or they do not cover enough years to be able to determine the maximum month with high confidence.*

- l.399: "anomalous": please define.

*not always anomalous = did not have high fluxes*

- l.402-403: "Rinne et al. (2020) also noted that methane emissions in four out of five Fennoscandian wetland sites were decreased in 2018 due to a decrease in water table levels. The summer months with high precipitation often resulted in high posterior emissions.": is this put here to suggest that the inversions are consistent with independent measurements? Shouldn't this be stated when presenting the inversions as a reference for comparison?

*Inversions are presented in the previous sentence: 'In the (inversion) posterior and especially in July 2018 the fluxes were decreased, possibly because of the decrease in soil water table level as June and July 2018 suffered from a lack of precipitation (see e.g. Peters et al., 2020)'*

*We added: Rinne et al. (2020) also noted that methane emissions in four out of five Fennoscandian wetland sites were decreased in 2018 due to a decrease in water table levels.* *This suggests that many wetlands suffered from drought, which could have affected the regional results.*

- l.404-405: "The year 2011 with observed high methane emissions from upland soil in northern

Fennoscandia (Lohila et al., 2016) did not stand out in posterior emissions": this suggests that the models do not well represent the high precipitation months. Why?

*It is possible that the signal observed by Lohila et al was not seen in the larger region of this study, but only in northern Fennoscandia.*

*We added this sentence in the text.*

- l.404-406: "The year 2011 with observed high methane emissions from upland soil in northern Fennoscandia (Lohila et al., 2016) did not stand out in posterior emissions, but large increases were assigned to high precipitation periods in e.g. August 2008, 2016 and late summer 2007": is this the same message as the sentence "The summer months with high precipitations..."?

*We changed the order of the sentences to make the message clearer.*

- l.406-407: "August was also the month of the average seasonal precipitation maximum, while the average temperature maximum was in July.": this sentence is the last in the paragraph. This suggests to the reader that it is very important. Nevertheless, it is actually only a description of the input data of temperature and precipitation, to state when their max is. Please, reorder the logical "progression" of the sections/paragraphs to provide the information in a logical progression.

*We changed the order of the two last paragraphs and sentences in the last paragraph, and added two summary sentences to the end (see text).*

**Conclusion**

The conclusion depends on how well the introduction presents the challenges tackled by the study. Therefore, in the current state of the text, seemingly trivial messages appear, e.g. that "it is important to study the overall responses of the emissions to air temperature and precipitations". The reader may winder that it is not already done and if this might due to the fact that models take only C stocks and water table depth into account. The conclusion will have to be re-written after taking into account the comments on all the previous sections.

- l.412-413: "agreed on the month of maximum emissions": this is not clear from the discussion.

*We clarified this in the last parapragh of the discussion.*

- l.413: "balanced": must be defined, see previous remarks.

*We added:* *both temperature and precipitation explain the variance of fluxes,*

- l.413: "significantly": see previous remarks.

*We changed to:* *was important*

- l.415: "to move emissions of both in posterior towards co-limitation of temperature and precipitation.": not clear, please reformulate.

*We reformulated the sentence:*

*… towards co-limitation of temperature and precipitation, i.e. to be sensitive to changes in both environmental variables.*

- l.415-417: "The set-up of different emission components (peatland emissions, mineral land fluxes) had a significant role in building up the response patterns": how is this not trivial? What is the link with the inversions

*We added the link with inversions*; …as they contribute to the total flux seen by inversions.

- l.419: "multi-year average response patterns": please explain.

*Where data from several years was used to determine the responses. We added this to the text.*

- l.420: "anomalous cases of severe droughts with significant water table drawdown in pristine peatlands": not obvious from the discussion. Pristine wetlands, "anomalous" and "significant" not defined in the text.

*We changed to: the cases of severe seasonal droughts leading to water table drawdown in peatlands*

- l.420-421: "corresponding reductions in methane emissions": Corresponding reductions are simulated by the models? The sentence seems to lack a verb.

*We changed to: which in turn leads to reductions…*

- l.421-423: "Depending on the model, wet mineral soil and inundated land emissions can modify the seasonality of methane emissions together with peatland emissions. Therefore, it is essential to pay more attention to the role of the individual emission components, their magnitude, annual cycle and spatial extent in different regions": how is this not trivial?

*This is not trivial as peatland emissions are considered to be the main component of methane emissions and wet mineral soils are often not considered to be as important. However, despite the lower emission per unit area, the regions emitting methane can be very large and therefore alter the seasonality of the regional emissions. Furthermore, the wet mineral lands are implemented in different ways in the models and can therefore produce different outcomes for the total flux.*

*We added this text in the conclusions*

- l.423-424: "how the fluxes should be scaled up from site to region": isn't it how people working on process model work? What does this study suggest they change in their work?

*When scaling up from site to region the area of the mineral soils, peatlands, lakes, rivers etc must be taken into account, as discussed in the introduction. In the boreal zone it is challenging to accurately define the extent of the different landcover classes that contribute to methane emissions, and when adding up different layers, there is a risk of erroneous or double counting of the emissions. Therefore, process models should use harmonized land cover datasets, and if the extents*

*of wet mineral lands and peatlands are simulated by the model, validate the estimate against satellite and /or ground based inventory datasets. The models should also use process-based descriptions for the mineral and peat soil fluxes to simulate the flux responses to climate drivers.*

*We added text about this in the conclusions:*

*…. To perform well in the Fennoscandia region, it is expected that a model will need to consider peatland and mineral land source and sink components of methane emissions and use up-to-date land cover description. If the peatland extent is simulated by the model, it needs to be validated against land cover data from bottom-up inventories. Similarly, for soil wetness or inundation, the model should be validated against satellite data wherever possible. The models should use process-based descriptions for both mineral and peat soil fluxes to simulate the flux responses to climate drivers.*

**Technical corrections**

- check the acronyms: they need to be fully explicit only at their first occurrence.

*We checked the acronyms*

- check the language: consistency between US/GB (s/z), accurate vocabulary outside of the strictly scientific (utilise/use) and the time of the verbs.

*We checked the language. For the time of verbs, the following guidance was applied: The past tense is mainly used to report the findings in the study and for literature review, while the present tense is used to interpret the results or refer to tables and figures, or for general rules and established knowledge.*

**Reviewer 2**

The authors investigated how simulated CH4 emissions from six ecosystem models respond to temperature and precipitation in Northern European regions. Then, simulated CH4 emissions from two ecosystem models exhibiting contrasting response patterns, and the ensemble mean of GCP atmospheric inversions, were used as priors of wetland emissions in an inversion. By doing this, they explored how the inversion changes the fluxes and alters the response of CH4 emissions to temperature and precipitation.

General comments:

Climate forcing used by the six ecosystem models are different. Previous studies have shown significant variations in process-based LSM outputs stemming from the choice of climate forcing. Four out of the six models were driven by CRUNCEPv7, one was driven by CRU-JRA and another one was driven by CRU-HARMONIE.

If my understanding is correct, temperature and precipitation data from CRU-JRA were applied to all six ecosystem models and the mean of GCP models, when studying the responses of simulated

CH4 emissions to temperature and precipitation (Fig. 1). Why not use the specific climate inputs that drive each model?

*To address this, we studied the effect of the climate input using results from uncoupled and coupled models (manuscript Fig. S1), where the driver data was either CRU-JRA for the uncoupled case or the ESM-specific climate model input for the coupled case. It would be expected that using coupled climate data as model input would lead to greater differences than simply using different uncoupled data sets, as all the uncoupled datasets (CRUNCEPv7, CRU-JRA and CRU-HARMONIE) are bias-corrected using precipitation and temperature measurements. The ESM-specific climate model data is not bias-corrected against such data. However, the response patterns were not significantly different (p<0.01). We are therefore confident that using CRU-JRA consistently throughout the paper produces robust results. To further test the impact of two different bias-corrected data sets, we created a figure (Fig. X1), where CRU-HARMONIE and CRU-JRA are plotted against JSBACH-H results, showing only small differences regarding the response of methane emissions to temperature and precipitation.*

[Figure]

*Figure X1: JSBACH ecosystem model results, plotted against left: CRU-HARMONIE ($R^2\_Temp$ =0.822, $R^2\_Prec$ = 0.054, $R^2\_TP$ = 0.834), right: CRU-JRA climate data ($R^2\_Temp$ =0.823, $R^2\_Prec$ = 0.0045, $R^2\_TP$ = 0.829).*

The representation of wetland area and inundation were various among models. Some models used a static and prescribed wetland area, some models used prescribed but time-varying wetland area, and other models dynamically simulated inundation and wetland area. It would be very useful to have a figure to show the wetland area used by each model.

*We added figures of wetland areas of two contrasting models. As the area of inundated land and wet mineral land varies over time, we provided exemplary maps for the most relevant time periods. The combined wetland area map including peatlands, inundated lands and wet mineral lands is quite different compared to the methane emission maps, because wet mineral land can cover wide land areas but their emissions per unit area are smaller than those from peatlands (See figures X2 - X5 below). Mineral soil was considered to be wet when the daily soil moisture, simulated for the 0.1 x 0.1 degree resolution grid cell, was high and there were methane emissions. Then the wetland fraction of the land area in that grid cell was 1.0, as the land area comprised of wet mineral land,*

*inundated land and peatland, all emitting methane. The wet fraction of mineral lands was calculated from the individual daily 0.1 x 0.1 degree resolution grid cells up-scaled and averaged over the month in question and over years 2000 - 2018.*

[Figure]

*Figure S5: Methane emitting land areas in JSBACH-H model, including a) peatlands, b) inundated lands and c) wet mineral lands average for the month of August over years 2000 – 2018.*

[Figure]

*Figure S6: Methane emitting land areas (including peatlands + inundated lands + wet mineral lands) in JSBACH-H model, for the month of a) July, b) August and c) September. The wetland fraction is calculated as an average for the month in question over years 2000 – 2018.*

[Figure]

*Figure S7: Methane emitting land areas in LPX-Bern model, including a) peatlands, b) inundated lands and c) wet mineral lands average for the month of August over years 2000 – 2018.*

[Figure]

*Figure S8: Methane emitting land areas (including peatlands + inundated lands + wet mineral lands) in LPX-Bern model, for the month of a) July, b) August and c) September. The wetland fraction is calculated as an average for the month in question over years 2000 – 2018.*

While the authors acknowledged the importance of wetland area in determining boreal regions CH4 emissions in both the introduction and the discussion section, the comparison of models' outputs and the analysis of precipitation and temperature responses didn't address the impact of wetland area.

*The discussion on the wetland area was embedded in the discussion of emissions from wet mineral soils and inundated land, as the magnitude of those is determined by the extent of the wetland area.*

*See e.g. section 'Model components and seasonal cycle' for LPX-Bern: 'The soil moisture and consequently the wet mineral land area peaked in autumn, and thus the wet mineral land emissions were at maximum in October in contrast to peatland emissions,….*

*To introduce the concept of wetland area, we added in the beginning of Chapter 3.2:* *…These components cover most of the land area in the study region. However, if we consider wetland area as land area which is wet enough to emit methane, only a fraction of land area is included in the wetland area. Wet mineral land area, inundated land area and peatland area with water table level close to the soil surface all contribute to the total wetland area, and total wetland emissions.*

Figure 7 shows substantial difference among models in terms of both the magnitude and the seasonal cycle of CH4 emissions. The influence of climate forcings and wetland area on these differences should be explicitly discussed.

*We added the following text on the impact of the wetland area, climate forcings and model features on the seasonal cycles:*

*Each model presented differences in seasonal cycle and peak month as they differed in wetland area and model process descriptions. On the contrary, the use of different climate-forcing data products or results from coupled climate simulations showed minor differences in methane emission responses (Figures S1 and S2).*

Though the authors showed both prior and posterior wetland CH4 emissions from the inversion model in Figure 6, as a reader not familiar with inversion models, I still find it challenging to grasp how the prior wetland CH4 emissions are adjusted. To achieve changes in both the magnitude and seasonal cycle of prior wetland CH4 emissions as shown in Figure 6, what other sources or sinks of CH4 have been altered?

*The inversion model used in the current work adjusts the emissions of CH4 every week. Thus, in addition to optimizing the magnitude of the emissions, the seasonal cycle is also adjusted. In addition to biospheric emissions inversions also optimize the anthropogenic emissions, but these have much less seasonal variability. Anthropogenic emissions are optimized separately but at the same time as natural emissions so that the total emissions produce modelled atmospheric concentrations that are consistent with concentration observations. There are also minor sources that are not optimized, from e.g. fires, termites and ocean, and the atmospheric sink which is calculated using prescribed data obtained from a chemistry model (e.g. OH fields).*

*We added some of these details to the description of CTE-CH4.*

Specific comments:

L266: explained CH4 emission -> CH4 emission

*We corrected the wording*

L338: Fig 1 should be Fig 2

*We corrected the number*

L392: to -> of?

*We corrected the wording*

Figure 3: A figure for upscaled flux observations could be added to show observation-based temperature and precipitation responses of wetland CH4 emissions.

*A figure was added (Fig. X6 below), using data from Olli Peltola et al 2019. There are not as many summer months of data as for the models, but the results suggest that the fluxes are correlated with both temperature and precipitation ($R^2\_Temp$ =0.83, $R^2\_Prec$ = 0.54, $R^2\_TP$ = 0.91).*

.

[Figure]

*Figure X6. Temperature and precipitation responses of upscaled methane emissions in Fennoscandia. Circles refer to monthly averages in May - September during years 2013-2014.*

Figure 6: The seasonal cycle of in situ atmospheric CH4 observations could be included in this figure to aid in understanding why the CTE-CH4 inversions shifted the monthly flux maximum towards August.

*The seasonal cycle of CH4 observations also has components other than biospheric emissions; anthropogenic emissions and atmospheric OH chemistry. The OH chemistry sink has a strong seasonal cycle with maximum impact in spring (see Fig X7 below)  and therefore the month of the summer maximum of the biospheric emissions is sometimes not clearly visible, and the visibility also depends on where the air masses have been transported. The summer maximum can often be seen as elevated concentrations in late summer, however it is minor compared to the winter maximum (which is caused by the combined effect of anthropogenic emissions continuing throughout the year, and a weak wintertime OH sink).   Therefore, adding a set of observations from different sites would not help much, but would make the figure more unclear for the reader. The inversion model solves for concentration changes due to anthropogenic and biospheric emissions, OH sink and atmospheric transport.*

[Figure]

*Figure X7. Methane concentrations measured at Pallas Sammaltunturi station in Finland.*

Figure 7: It is difficult to distinguish lines for LPX-Bern and LPJ-GUESS, lines for the four GCPs, due to similar colors.

*We added different markers for each line.*